# CASTLE-IN-THE-AIR: EVALUATING MLLM VISUAL ABILITIES ON HUMAN COGNITIVE BENCHMARKS

## ABSTRACT

Despite significant progress on popular multimodal benchmarks, state-of-the-art Multimodal Large Language Models (MLLMs) continue to struggle with basic visual reasoning tasks that are trivially solved by humans, such as recognizing abstract patterns or identifying spatial relationships. Such deficiencies undermine their efficacy and robustness, rendering high-level downstream applications (*e.g.*, embodied AI) infeasible. To systematically investigate this gap, we introduce **VISFACTOR**, a benchmark that digitizes 20 vision-centric subtests from FRCT, a well-established cognitive psychology assessment, including four domains of human visual cognition: (1) Visualization and Spatial Processing, (2) Perceptual and Closure, (3) Memory, and (4) Reasoning. Furthermore, we leverage parametric generation to automatically construct unlimited test cases with controllable difficulty for applicable subtests. Using **VISFACTOR**, we evaluate 23 frontier MLLMs, including both proprietary (GPT, Gemini, *etc.*) and open-source models (LLaMA-3.2, Qwen2.5-VL, *etc.*). The best-performing model achieves a score of only 30.17%, with consistent failures on tasks such as mental rotation, spatial relation inference, and figure–ground discrimination—regardless of model size or prompting strategy. These findings suggest that performance improvements on existing general benchmarks might be *castles in the air* instead of mastery of human-like visual cognition, challenging the assumption that large-scale pretraining naturally induces gestalt-like perceptual capabilities. The dataset and evaluation toolkit will be made publicly available upon publication.

## 1 INTRODUCTION

Multimodal Large Language Models (MLLMs) have rapidly advanced the state of multimodal artificial intelligence, delivering impressive results in text recognition (Liu et al., 2024b; Chen et al., 2025), mathematical reasoning (Yang et al., 2024; Peng et al., 2024), and even clinical decision support (Azad et al., 2023; Buckley et al., 2023; Ye et al., 2024). On holistic benchmarks such as MMBench (Liu et al., 2024a), frontier models like Gemini-2.5-Pro (Kavukcuoglu, 2025) have reached nearly *90%* accuracy—results that have fueled optimism that large-scale pretraining may already confer near-human visual cognition. However, closer inspection tells a different story. Targeted studies show MLLMs still fail on visual reasoning problems that human novices solve effortlessly (Fu et al., 2024). For example, Ramakrishnan et al. (2025) reports near-random accuracy on mental rotation test and maze completion test. This paradox highlights a key limitation in current evaluation paradigms: most benchmarks primarily emphasize aggregate scores on downstream tasks, largely neglecting to probe the *foundational* visual faculties that underlie human reasoning.

Human vision develops hierarchically: low-level perceptual skills—figure–ground segregation, object permanence, spatial scanning—serve as scaffolds for higher-order reasoning. This raises a critical question: do MLLMs truly possess the human-like cognitive visual abilities required for complex tasks? In cognitive psychology, researchers decompose vision into latent factors that can be measured independently. The *Factor-Referenced Cognitive Test* (FRCT) battery (Ekstrom & Harman, 1976) operationalizes this by mapping psychometric factors to narrowly defined subtests. In contrast to omnibus IQ scales, the FRCT delivers a fine-grained cognitive profile, making it ideal for diagnosing the precise visual capacities an MLLM truly possesses.

We introduce **VISFACTOR**, which for the first time, adapts 20 vision-centric FRCT subtests into an automated, multimodal benchmark specifically designed for MLLMs. **VISFACTOR** spans four critical cognitive domains: (1) visualization and spatial processing, (2) perceptual and closure, (3) memory, and (4) reasoning (Fig. 5). Prior multimodal benchmarks (Ramakrishnan et al., 2025; Fu et al., 2024) often rely on multiple-choice ($1/N$ chance with N choices) or True-or-False ($1/2$ chance) question formats. Such formats introduce significant random guessing opportunities, often failing to reveal true performance disparities between models. To deliver a more rigorous evaluation, we generate at least four rule-based variants for every question and deliberately diversify the correct-answer distribution.[1] This design reduces the overall chance-level accuracy to *2.9%*, ensuring that any success on **VISFACTOR** reflects genuine visual reasoning rather than lucky guesses.

We evaluated 23 frontier MLLMs spanning major families, including GPT (Hurst et al., 2024), Gemini (Kavukcuoglu, 2025), Claude (Anthropic, 2025b), LLaMA (Meta, 2024), Qwen (Bai et al., 2025), and SEED (ByteDance, 2025). Despite advanced prompting strategies such as Chain-of-Thought (CoT) (Kojima et al., 2022; Wei et al., 2022), the best model scores only *30.17%* of accuracy. Systematic failures consistently manifested in core visual tasks such as mental rotation, spatial relation inference, and figure–ground discrimination, irrespective of model size or architecture.

The original FRCT has a finite item set, which poses a risk of direct overfitting by future models. To future-proof **VISFACTOR**, we focus on the subtests where current models demonstrate the weakest performance and implement a parametric generator. The generator produces an unlimited supply of difficulty-controlled instances that faithfully adhere to the FRCT style. Key question parameters (*e.g.*, rotation angle, occlusion level, grid size) can be precisely modulated to create graduated test suites, enabling robust tracking without saturating the benchmark. Our contributions are as follows:

1. We present the first benchmark that grounds MLLM assessment directly to human cognitive factors, bringing psychometric rigor to multimodal evaluation.
2. Implemented with VLMEvalKit (Duan et al., 2024), we digitize FRCT vision items, devise variants, and synthesize controllable-difficulty items for harder tests and model reinforcement learning.
3. We benchmark 23 state-of-the-art MLLMs, offering a panoramic view of current capabilities and pinpointing cognitive gaps that chart a roadmap for future research.

## 2 VISFACTOR DESIGN AND IMPLEMENTATION

This section introduces how we select tests from FRCT (§2.1), how to fit the tests to MLLMs (§2.2-§2.3), and how we generate more difficulty-controllable test cases (§2.4).

### 2.1 TEST SELECTION AND JUSTIFICATION

The original FRCT battery comprises 72 subtests. We exclude those that cannot be assessed with a vision–language interface whose output is text only: (1) **Image-production tasks** (4 subtests): Figural Fluency (FF1–FF3) and Spatial Scanning (SS1) ask participants to draw or trace; this is incompatible with text-only output. (2) **Speech-dependent tasks** (3 subtests): Memory Span (MS1–MS3) require subjects to write down what they hear and therefore probe speech-to-text rather than visual cognition. In the remaining 65 subtests, 45 of them can be completed with pure text input. Those demanding visual reasoning but accept text answers form our benchmark, **VISFACTOR**. The 20 subtests cover 10 FRCT factors: Closure Flexibility (CF), Closure Speed (CS), Induction (I), Associative Memory (MA), Visual Memory (MV), Perceptual Speed (P), Logical Reasoning (RL), Spatial Orientation (S), Spatial Scanning (SS), and Visualization (VZ). Figure 1 shows example questions and answers of each subtest. Dataset statistics are included in Fig. 5 and Table 6 in §6.

### 2.2 DIGITIZATION AND PROMPT DESIGN

**(1) Instructions.** Directly feeding the human-oriented FRCT instructions to MLLMs prove verbose and occasionally ambiguous. We therefore ask GPT-4o and Gemini-2.5-Flash to summarize each instruction set to its minimal, MLLM-friendly form. A human annotator reconcile the two summaries with the originals, producing a concise final prompt for every subtest. **(2) Questions and**

---

[1]For instance, multiple-choice keys are not always "A", nor are "Yes/No" items disproportionately "Yes".

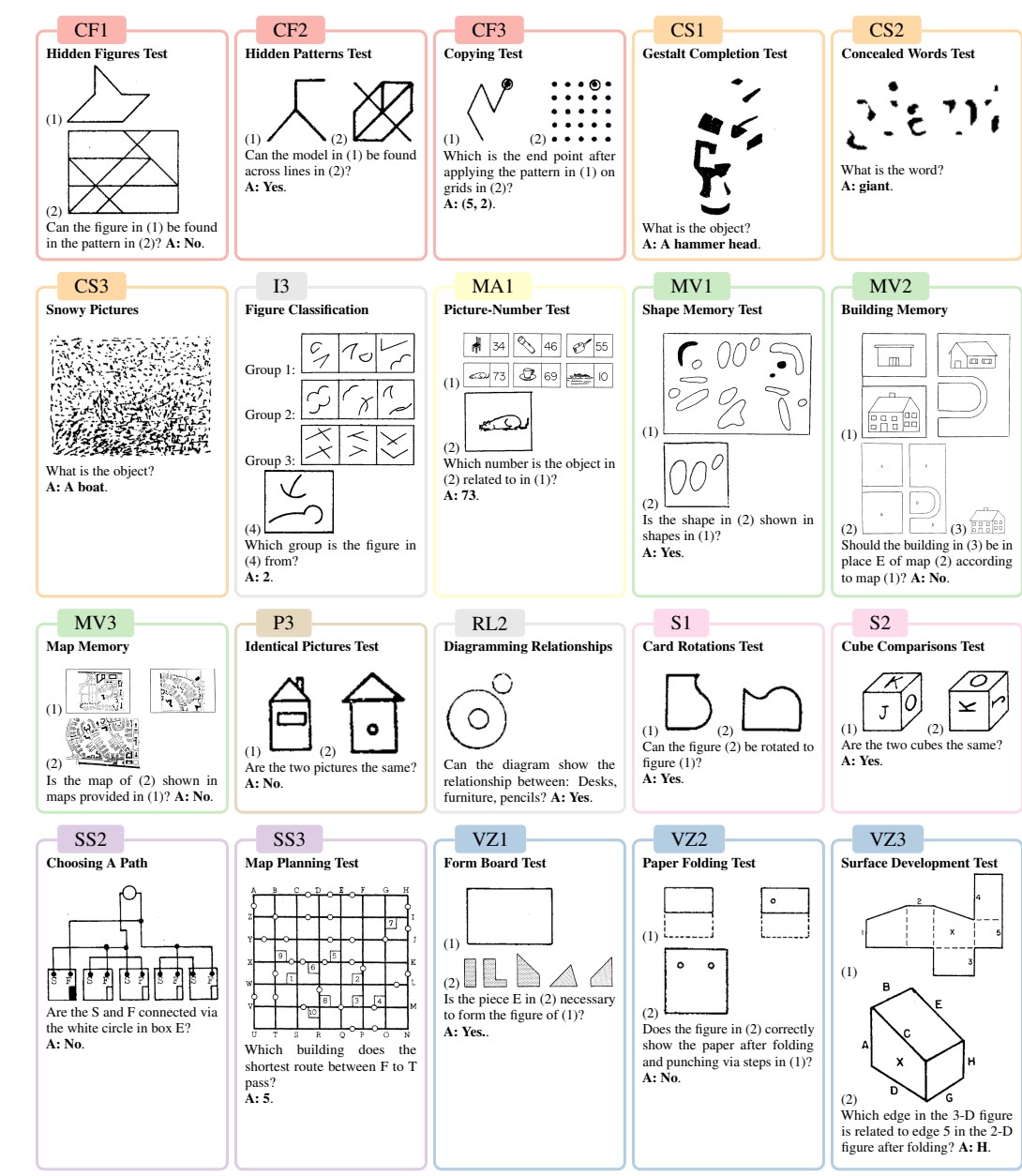

Figure 1: **VISFACTOR** comprises 20 vision-centric cognitive subtests. Each task is designed to isolate core factors of human visual cognition, covering 10 distinct factors in total. The subtests are converted into either yes/no questions or fill-in-the-blank questions according to §2.3. Example stimuli, questions, and ground-truth answers are shown for each task.

**Answers.** All images are captured at $300\,\mathrm{dpi}$ and cropped to the region containing only the task stimuli (no additional texts). Ground-truth answers are extracted verbatim from the FRCT manuals.

## 2.3 REDUCING CHANCE-LEVEL ACCURACY

To prevent inflated scores from lucky guesses, we modify test formats as follows, except CF3 (25-way), MA1 (21-way) and all fill-in-the-blank subtests (CS1-CS3) that already exhibit $\leq 5\%$ random success. The average random guessing performance is reduced from $22.47\%$ to $2.89\%$, with no single test exceeding $6.25\%$.

1. **Decomposed multiple choice**: For seven subtests with five options (CF1, MV2, P3, RL2, SS2, VZ1, VZ2), we pose *one yes/no query per option* and require the model to answer *all* correctly for credit. Chance accuracy thus drops from 25% to $(0.5)^5 \approx 3.13\%$.

2. **Grouped-consistency items**: Three subtests repeatedly probe the same latent feature across a small cluster of items. We aggregate each cluster and award credit only if *all* constituent items are correct. This applies to: *(i)* CF2 Hidden Patterns Test—400 binary items grouped into 80 sets of five; chance $(0.5)^5 \approx 3.13\%$. *(ii)* I3 Figure Classification—eight figures to be classified into two or three groups; chance $\approx 0.23\%$. *(iii)* S1 Card Rotation Test—eight judgments of the same card; chance $(0.5)^8 \approx 0.39\%$.

3. **Symmetry variants**: MV1, MV3 and S2 originally ask whether figure A matches figure B. We generate three variants per item—"A differs from B", "B matches A", "B differs from A"—so that "yes" and "no" answers are balanced, preventing easy success by models that consistently answer yes or no. The probability of guessing all three correctly by chance is $(0.5)^4 = 6.25\%$.

4. **Specialized rewrites**: *(i)* SS3 (Map Planning Test). Each item asks participants to find the building number that the shortest path between a *start* and an *end* point passes in a map. Exchanging start and end leaves the correct answer unchanged. We therefore require the model to answer *both* directions correctly, lowering chance from 10% to 1%. *(ii)* VZ3 (Surface Development Test). Each item asks: which 3-D edge corresponds to the marked 2-D edge after folding? Since multiple 2-D edges may map to the same 3-D edge, simply swapping the query direction (asking which 2-D edge matches a given 3-D edge) would introduce one-to-many ambiguity and ill-defined ground truth. Therefore, we add additional questions asking whether a pair of 2-D and 3-D edges are the same, resulting in all "yes" ground truth. To create "no" pairs, we generate questions with cyclic-permuted 3-D edge labels (*e.g.*, $A \to B \to C \to D \to E \to A$). MLLMs receive credit only if they correctly answer the fill-in-the-blank question and both yes/no questions; chance $14.6/4 = 3.65\%$.

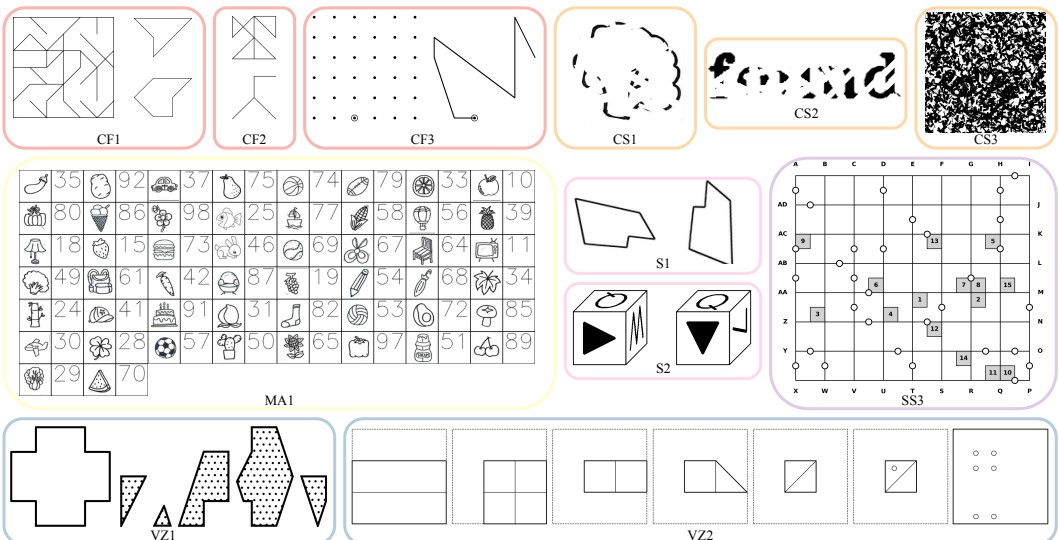

Figure 2: Samples of our generated images. We can dynamically adjust test difficulties in **VISFACTOR**. For example, the grid size of CF3 is changed to $6 \times 6$ instead of $5 \times 5$.

## 2.4 SYNTHETIC AUGMENTATION

We implement parametric generation for a subset—CF1–CF3, CS1–CS3, MA1, S1–S2, SS3, VZ1–VZ2. Fig. 2 illustrates sample questions generated by our algorithms. To guarantee correctness, we carefully design algorithms that produce valid question–answer pairs. For example, in the S2 Cube Comparison Test, we design an algorithm to determine whether two cubes—represented by three characters denoting the upper, front, and right faces, along with their rotation angles—are identical. In the VZ2 Paper Folding Test, our method randomly selects symmetry axes, folds the

figure, punches a hole, and then unfolds it in reverse order to obtain the final answer. Due to space limit, the details of our algorithms are included in §C of the appendix.

## 3 EXPERIMENTS

### 3.1 SETTINGS

**Models.** We evaluate 23 models: GPT-4o (Hurst et al., 2024), GPT-4o-Mini (OpenAI, 2024), GPT-4.1 (OpenAI, 2025a), GPT-5-Mini (OpenAI, 2025b), GPT-5.1 (with different reasoning efforts) (OpenAI, 2025c), Gemini-2.5-(Pro, Flash) (Kavukcuoglu, 2025), Claude-Sonnet-(3.5 (Anthropic, 2024), 3.7 (Anthropic, 2025a), 4 (Anthropic, 2025b)), Qwen-2-VL (Wang et al., 2024a), Qwen-2.5-VL-(32B, 72B) (Bai et al., 2025), Qwen-3-VL-Plus (Team, 2025), Qwen-VL-Max (Team, 2024), Seed-1.5-VL (Guo et al., 2025), Seed-1.6 (ByteDance, 2025), Moonshot-V1-128K-Vision-Preview (MoonshotAI, 2025), LLaMA-3.2-Vision-(11B, 90B) (Meta, 2024), o1 (Jaech et al., 2024), o3 (OpenAI, 2025d), and o4-Mini (OpenAI, 2025d).

**Hyper-parameters and Prompts.** We set the temperature to 0 for all models, except Qwen (minimum temperature 0.01) and LLaMA-3.2 (temperature 0.6). For Qwen, Top-P is set to 0.001; for LLaMA-3.2, Top-P is set to 0.9. The thinking budget is configured as *high* for Gemini-2.5, GPT-5.1, and o-series models. Greedy decoding is used as the default sampling strategy. All models are accessed via their official APIs, except LLaMA-3.2, which is run locally. In our implementation, the retry count is set to 3, allowing each case up to three retries before being marked as a failure. All test cases are conducted in a zero-shot setting. The exact prompts are provided in §B of the appendix.

**Evaluation Criteria.** We adopt a unified and fully specified evaluation protocol across all task types. For yes-no questions, model outputs are normalized and matched to the gold label using the sets $\{t, y, 1, \texttt{true}, \texttt{yes}\}$ for the True class and $\{f, n, 0, \texttt{false}, \texttt{no}\}$ for the False class. Multiple-choice questions are evaluated by directly matching the predicted option letter (A/B/C/D). For numeric fill-in-the-blank problems, we require exact numeric matching. For CS1 and CS3, multiple acceptable ground-truth answers are provided; a prediction is counted as correct if its normalized form matches any valid answer variant. CS2 uses strict exact-match evaluation. This unified criterion ensures consistency across task formulations while accommodating the natural variability of certain cognitive tasks.

### 3.2 RESULTS ON ORIGINAL TESTS

**Most existing models perform poorly on the VISFACTOR benchmark.** Among the 23 evaluated frontier models, *GPT-5.1* achieves the highest overall score, but only reaches *30.17%* out of 100. Even when aggregating the best-performing models across individual subtests, the combined score is just *40.0%*. Models generally perform relatively better on memorization tasks (MA1, MV1–MV3), indicating a strong ability to attend to relevant context in the input (a detailed study is included in §4.1). A breakdown of top-performing models by subtest reveals distinct strengths: (i) OpenAI's o-series models excel at reasoning tasks (I3, RL2). They also perform best on CF1–CF3 and CS1, demonstrating superior recognition of lines and edges. (ii) Google's Gemini leads on P3 and VZ2, particularly excelling at VZ2, which requires precise spatial localization to identify holes in paper. (iii) Qwen leads on SS2, VZ1, and VZ3, indicating strong mental imagery capabilities for shape splicing and folding. (iv) Claude performs best on CS2, MV1, MV3, S2, and SS3. (v) Seed achieves the top score on CS3 and MV2.

**Model size and recency do not guarantee superior performance.** For example, Qwen-2.5-32B outperforms Qwen-2.5-72B, and Qwen-2-72B also surpasses Qwen-2.5-72B. Similarly, Claude-3.7 outperforms Claude-4, and Seed-1.5 exceeds Seed-1.6. While there are exceptions—such as GPT-4o outperforming GPT-4o-Mini, and o3 surpassing o1—performance on VISFACTOR shows no consistent correlation with model scale or version. These results suggest that core visual capabilities may be underemphasized in current model development pipelines.

**Reasoning models gain from longer CoT, but non-reasoning models do not.** We evaluate the effect of CoT prompting across three GPT models (GPT-4.1-2025-04-14, GPT-4o-2024-11-20, GPT-4o-Mini-2024-07-18). While CoT provides some improvements, the gains in overall performance

Table 1: The performance of 23 models on **VISFACTOR**. The bottom row shows the highest scores achieved by any model, while the rightmost column shows the total score. Darker scores show higher scores. The best model is GPT-5.1.

| | CF1 | CF2 | CF3 | CS1 | CS2 | CS3 | I3 | MA1 | MV1 | MV2 | MV3 | P3 | RL2 | S1 | S2 | SS2 | SS3 | VZ1 | VZ2 | VZ3 | Total Score |
|---|---|---|---|---|---|---|---|---|---|---|---|---|---|---|---|---|---|---|---|---|---|
| Claude-3.5-Sonnet-2024-10-22 | 0.0 | 1.2 | 6.2 | 10.0 | 14.0 | 4.2 | 7.1 | 100.0 | 31.2 | 4.2 | 70.8 | 41.7 | 20.0 | 0.0 | 52.4 | 6.2 | 20.0 | 2.1 | 0.0 | 10.0 | 20.1 |
| Claude-3.7-Sonnet | 6.2 | 1.2 | 1.6 | 5.0 | 18.0 | 4.2 | 14.3 | 100.0 | 53.1 | 20.8 | 95.8 | 37.5 | 43.3 | 0.0 | 40.5 | 9.4 | 20.0 | 14.6 | 0.0 | 18.3 | 25.2 |
| Claude-4-Sonnet | 3.1 | 8.8 | 9.4 | 0.0 | 10.0 | 4.2 | 7.1 | 100.0 | 21.9 | 8.3 | 45.8 | 40.6 | 33.3 | 0.0 | 21.4 | 0.0 | 25.0 | 8.3 | 0.0 | 1.7 | 17.4 |
| GPT-4.1-2025-04-14 | 0.0 | 7.5 | 0.0 | 10.0 | 10.0 | 8.3 | 17.9 | 100.0 | 53.1 | 8.3 | 66.7 | 49.0 | 23.3 | 0.0 | 28.6 | 0.0 | 17.5 | 16.7 | 5.0 | 5.0 | 21.3 |
| GPT-4o-2024-11-20 | 0.0 | 15.0 | 6.2 | 15.0 | 8.0 | 8.3 | 21.4 | 100.0 | 31.2 | 0.0 | 62.5 | 69.8 | 16.7 | 0.0 | 26.2 | 3.1 | 20.0 | 18.8 | 0.0 | 5.0 | 21.4 |
| GPT-4o-Mini-2024-07-18 | 6.2 | 1.2 | 4.7 | 20.0 | 4.0 | 8.3 | 10.7 | 100.0 | 6.2 | 0.0 | 54.2 | 32.3 | 3.3 | 0.0 | 42.9 | 3.1 | 17.5 | 12.5 | 0.0 | 0.0 | 16.4 |
| GPT-5-Mini-2025-08-07 | 0.0 | 11.2 | 17.2 | 10.0 | 10.0 | 4.2 | 28.6 | 100.0 | 9.4 | 20.8 | 50.0 | 45.8 | 90.0 | 0.0 | 31.0 | 15.6 | 17.5 | 4.2 | 5.0 | 15.0 | 24.3 |
| GPT-5.1-2025-11-13-High | 0.0 | 18.8 | 25.0 | 20.0 | 6.0 | 16.7 | 42.9 | 100.0 | 43.8 | 8.3 | 75.0 | 38.5 | 96.7 | 5.0 | 38.1 | 12.5 | 12.5 | 2.1 | 25.0 | 16.7 | 30.2 |
| GPT-5.1-2025-11-13-Low | 3.1 | 10.0 | 18.8 | 15.0 | 10.0 | 12.5 | 35.7 | 100.0 | 50.0 | 12.5 | 79.2 | 34.4 | 83.3 | 0.0 | 14.3 | 3.1 | 15.0 | 4.2 | 10.0 | 15.0 | 26.3 |
| GPT-5.1-2025-11-13-None | 3.1 | 13.8 | 3.1 | 20.0 | 16.0 | 8.3 | 17.9 | 100.0 | 46.9 | 0.0 | 83.3 | 32.3 | 3.3 | 0.0 | 28.6 | 0.0 | 10.0 | 2.1 | 0.0 | 8.3 | 19.9 |
| Gemini-2.5-Flash | 0.0 | 8.8 | 9.4 | 10.0 | 0.0 | 8.3 | 21.4 | 97.6 | 25.0 | 8.3 | 41.7 | 54.2 | 50.0 | 0.0 | 11.9 | 0.0 | 0.0 | 0.0 | 5.0 | 0.0 | 17.6 |
| Gemini-2.5-Pro | 0.0 | 13.8 | 4.7 | 20.0 | 6.0 | 12.5 | 28.6 | 100.0 | 3.1 | 0.0 | 0.0 | 77.1 | 13.3 | 0.0 | 2.4 | 3.1 | 7.5 | 18.8 | 35.0 | 1.7 | 17.4 |
| LLaMA-3.2-11B-Vision-Instruct | 0.0 | 7.5 | 3.1 | 5.0 | 6.0 | 0.0 | 0.0 | 0.0 | 0.0 | 0.0 | 4.2 | 3.1 | 0.0 | 0.0 | 9.5 | 3.1 | 2.5 | 4.2 | 0.0 | 0.0 | 2.4 |
| LLaMA-3.2-90B-Vision-Instruct | 9.4 | 0.0 | 10.9 | 0.0 | 4.0 | 8.3 | 3.6 | 0.0 | 12.5 | 0.0 | 8.3 | 7.3 | 0.0 | 0.0 | 0.0 | 0.0 | 17.5 | 0.0 | 0.0 | 0.0 | 4.1 |
| Moonshot-v1-128K-Vision-Preview | 0.0 | 0.0 | 1.6 | 0.0 | 2.0 | 4.2 | 7.1 | 69.0 | 12.5 | 0.0 | 25.0 | 40.6 | 0.0 | 0.0 | 19.0 | 0.0 | 7.5 | 2.1 | 0.0 | 0.0 | 9.5 |
| Qwen-2-VL-72B-Instruct | 0.0 | 1.2 | 9.4 | 0.0 | 6.0 | 8.3 | 3.6 | 95.2 | 18.8 | 0.0 | 58.3 | 40.6 | 0.0 | 0.0 | 26.2 | 0.0 | 22.5 | 22.9 | 0.0 | 16.7 | 16.5 |
| Qwen-2.5-VL-32B-Instruct | 9.4 | 8.8 | 0.0 | 0.0 | 10.0 | 0.0 | 3.6 | 92.9 | 21.9 | 4.2 | 54.2 | 41.7 | 0.0 | 0.0 | 2.4 | 0.0 | 10.0 | 0.0 | 0.0 | 6.7 | 13.3 |
| Qwen-2.5-VL-72B-Instruct | 9.4 | 2.5 | 9.4 | 5.0 | 2.0 | 4.2 | 3.6 | 95.2 | 0.0 | 0.0 | 0.0 | 53.1 | 0.0 | 0.0 | 0.0 | 0.0 | 12.5 | 20.8 | 0.0 | 0.0 | 10.9 |
| Qwen-3-VL-Plus-2025-09-23 | 3.1 | 8.8 | 4.7 | 10.0 | 12.0 | 4.2 | 14.3 | 100.0 | 34.4 | 8.3 | 75.0 | 67.7 | 6.7 | 0.0 | 35.7 | 0.0 | 25.0 | 6.2 | 0.0 | 16.7 | 21.6 |
| Qwen-VL-Max-2025-04-08 | 0.0 | 8.8 | 7.8 | 5.0 | 10.0 | 0.0 | 14.3 | 100.0 | 28.1 | 4.2 | 54.2 | 58.3 | 6.7 | 0.0 | 50.0 | 12.5 | 15.0 | 20.8 | 5.0 | 23.3 | 21.2 |
| Seed-1.5-VL | 0.0 | 1.2 | 6.2 | 10.0 | 6.0 | 12.5 | 14.3 | 100.0 | 50.0 | 41.7 | 79.2 | 10.4 | 53.3 | 0.0 | 47.6 | 3.1 | 15.0 | 2.1 | 5.0 | 16.7 | 23.7 |
| Seed-1.6-Thinking | 3.1 | 3.8 | 12.5 | 15.0 | 0.0 | 0.0 | 10.7 | 100.0 | 18.8 | 16.7 | 66.7 | 54.2 | 53.3 | 0.0 | 11.9 | 12.5 | 22.5 | 4.2 | 5.0 | 18.3 | 21.5 |
| o1-2024-12-17 | 6.2 | 1.2 | 9.4 | 20.0 | 10.0 | 12.5 | 35.7 | 92.9 | 37.5 | 4.2 | 62.5 | 4.2 | 90.0 | 0.0 | 16.7 | 0.0 | 7.5 | 0.0 | 0.0 | 5.0 | 20.8 |
| o3-2025-04-16 | 0.0 | 16.2 | 18.8 | 25.0 | 2.0 | 8.3 | 42.9 | 85.7 | 21.9 | 8.3 | 62.5 | 28.1 | 90.0 | 0.0 | 31.0 | 12.5 | 2.5 | 10.4 | 5.0 | 15.0 | 24.3 |
| o4-Mini-2025-04-16 | 9.4 | 2.5 | 18.8 | 15.0 | 8.0 | 8.3 | 14.3 | 97.6 | 28.1 | 16.7 | 66.7 | 37.5 | 90.0 | 0.0 | 31.0 | 0.0 | 5.0 | 2.1 | 5.0 | 8.3 | 23.2 |
| GPT-4.1-2025-04-14-CoT | 6.2 | 7.5 | 6.2 | 10.0 | 8.0 | 4.2 | 25.0 | 100.0 | 18.8 | 8.3 | 54.2 | 49.0 | 63.3 | 0.0 | 47.6 | 0.0 | 15.0 | 2.1 | 0.0 | 11.7 | 21.9 |
| GPT-4o-2024-11-20-CoT | 0.0 | 1.2 | 0.0 | 20.0 | 6.0 | 16.7 | 25.0 | 100.0 | 50.0 | 12.5 | 54.2 | 47.9 | 3.3 | 0.0 | 47.6 | 0.0 | 10.0 | 12.5 | 5.0 | 13.3 | 21.3 |
| GPT-4o-Mini-2024-07-18-CoT | 3.1 | 1.2 | 3.1 | 20.0 | 2.0 | 12.5 | 17.9 | 100.0 | 12.5 | 0.0 | 75.0 | 29.2 | 3.3 | 0.0 | 40.5 | 12.5 | 0.0 | 16.7 | 0.0 | 6.7 | 17.8 |
| Model Max | 9.4 | 18.8 | 25.0 | 25.0 | 18.0 | 16.7 | 42.9 | 100.0 | 53.1 | 41.7 | 95.8 | 77.1 | 96.7 | 5.0 | 52.4 | 15.6 | 25.0 | 22.9 | 35.0 | 23.3 | 40.0 |

are marginal. A correlation analysis between CoT token count and accuracy shows negative Pearson correlations of -0.18, -0.28, and -0.35, respectively. This analysis indicates that longer CoT often reflects uncertainty rather than improved reasoning, and CoT length is not a reliable proxy for reasoning quality. This aligns with recent findings showing that CoT does not universally enhance model performance; in fact, certain cognitive tasks may exhibit degraded performance with CoT (Liu et al., 2025a). Specifically, we observe declines in performance on perceptual and closure tasks (P3, CS2) and spatial visualization tasks (SS3, VZ1). Conversely, CoT consistently improves performance on reasoning tasks such as I3 and RL2, consistent with prior results (Sprague et al., 2025). Table 1 shows that GPT-5.1-High outperforms the low- and none-reasoning variants. This supports the hypothesis that dedicated reasoning models benefit from extended CoT, whereas non-reasoning models which lack specialized training gain little or no improvement from longer chains.

**The "Middle Score Anomaly" (Babaiee et al., 2025) is also observed in our VISFACTOR.** This phenomenon refers to models unexpectedly achieving intermediate performance—neither random nor near-perfect—on tasks that are extremely easy for humans. For instance, the Identical Pictures Test (P3) simply requires determining whether two images depict the same object. Humans can either solve this task almost perfectly or fail entirely (*i.e.*, perform at chance level if they lack the necessary perceptual ability). It would be highly unusual for a human to achieve, say, 70% accuracy on this task—suggesting partial understanding but inexplicable failures. However, we observe that most models obtain 30–50% accuracy on P3, while random guessing yields only 3.13%. We interpret this as evidence that current models lack genuine reasoning capabilities, at least in the context of the tasks presented in **VISFACTOR**. Our further failure analysis (see Appendix 4 due to space limit) reveals that the apparent strengths of current MLLMs often stem from concept-level recognition rather than genuine cognitive visual processing.

**Temperatures bring marginal influence.** To assess model robustness against temperatures, we evaluate temperatures 0.5 and 1.0 for three models: GPT-4.1-2025-04-14, GPT-4o-2024-11-20, and GPT-4o-Mini-2024-07-18. As shown in Table 2, the overall performance fluctuates only marginally across temperature settings, and the total score remains stable, indicating that our conclusions are not sensitive to the choice of decoding temperature.

Table 2: The performance of GPT-4.1-2025-04-14, GPT-4o-2024-11-20, and GPT-4o-Mini-2024-07-18 in **VISFACTOR** using different temperatures of $\{0.0, 0.5, 1.0\}$.

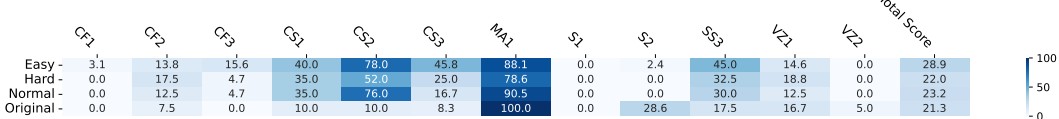

| | CF1 | CF2 | CF3 | CS1 | CS2 | CS3 | I3 | MA1 | MV1 | MV2 | MV3 | P3 | RL2 | S1 | S2 | SS2 | SS3 | VZ1 | VZ2 | VZ3 | Total Score |
|---|---|---|---|---|---|---|---|---|---|---|---|---|---|---|---|---|---|---|---|---|---|
| GPT-4.1-2025-04-14-T0.0 | 0.0 | 7.5 | 0.0 | 10.0 | 10.0 | 8.3 | 17.9 | 100.0 | 53.1 | 8.3 | 66.7 | 49.0 | 23.3 | 0.0 | 28.6 | 0.0 | 17.5 | 16.7 | 5.0 | 5.0 | 21.3 |
| GPT-4.1-2025-04-14-T0.5 | 3.1 | 13.8 | 1.6 | 10.0 | 12.0 | 8.3 | 21.4 | 100.0 | 56.2 | 12.5 | 70.8 | 50.0 | 23.3 | 0.0 | 26.2 | 0.0 | 20.0 | 8.3 | 0.0 | 5.0 | 22.1 |
| GPT-4.1-2025-04-14-T1.0 | 0.0 | 13.8 | 4.7 | 10.0 | 10.0 | 8.3 | 17.9 | 100.0 | 53.1 | 8.3 | 70.8 | 47.9 | 20.0 | 0.0 | 33.3 | 0.0 | 17.5 | 10.4 | 0.0 | 5.0 | 21.6 |
| GPT-4o-2024-11-20-T0.0 | 0.0 | 15.0 | 6.2 | 15.0 | 8.0 | 8.3 | 21.4 | 100.0 | 31.2 | 0.0 | 62.5 | 69.8 | 16.7 | 0.0 | 26.2 | 3.1 | 20.0 | 18.8 | 0.0 | 5.0 | 21.4 |
| GPT-4o-2024-11-20-T0.5 | 3.1 | 20.0 | 1.6 | 10.0 | 12.0 | 8.3 | 25.0 | 100.0 | 34.4 | 8.3 | 66.7 | 69.8 | 0.0 | 0.0 | 26.2 | 0.0 | 12.5 | 18.8 | 0.0 | 1.7 | 20.9 |
| GPT-4o-2024-11-20-T1.0 | 0.0 | 18.8 | 1.6 | 10.0 | 10.0 | 4.2 | 17.9 | 100.0 | 34.4 | 0.0 | 62.5 | 64.6 | 13.3 | 0.0 | 23.8 | 0.0 | 27.5 | 20.8 | 0.0 | 1.7 | 20.5 |
| GPT-4o-Mini-2024-07-18-T0.0 | 6.2 | 1.2 | 4.7 | 20.0 | 4.0 | 8.3 | 10.7 | 100.0 | 6.2 | 0.0 | 54.2 | 32.3 | 3.3 | 0.0 | 42.9 | 3.1 | 17.5 | 12.5 | 0.0 | 0.0 | 16.4 |
| GPT-4o-Mini-2024-07-18-T0.5 | 3.1 | 1.2 | 6.2 | 20.0 | 4.0 | 8.3 | 10.7 | 100.0 | 3.1 | 0.0 | 50.0 | 30.2 | 3.3 | 0.0 | 38.1 | 0.0 | 15.0 | 4.2 | 0.0 | 1.7 | 15.0 |
| GPT-4o-Mini-2024-07-18-T1.0 | 3.1 | 1.2 | 9.4 | 25.0 | 6.0 | 8.3 | 14.3 | 97.6 | 15.6 | 8.3 | 41.7 | 32.3 | 6.7 | 0.0 | 33.3 | 3.1 | 10.0 | 4.2 | 5.0 | 5.0 | 16.5 |

Table 3: The performance of the GPT-4.1 model on the generated subsets in **VISFACTOR**. The "Original" row reports performance on the original FRCT questions. The "Normal" row uses the same configuration as the original questions. The "Easy" and "Hard" rows correspond to questions that are modified to be easier and more difficult, respectively.

| | CF1 | CF2 | CF3 | CS1 | CS2 | CS3 | MA1 | S1 | S2 | SS3 | VZ1 | VZ2 | Total Score |
|---|---|---|---|---|---|---|---|---|---|---|---|---|---|
| Easy | 3.1 | 13.8 | 15.6 | 40.0 | 78.0 | 45.8 | 88.1 | 0.0 | 2.4 | 45.0 | 14.6 | 0.0 | 28.9 |
| Hard | 0.0 | 17.5 | 4.7 | 35.0 | 52.0 | 25.0 | 78.6 | 0.0 | 0.0 | 32.5 | 18.8 | 0.0 | 22.0 |
| Normal | 0.0 | 12.5 | 4.7 | 35.0 | 76.0 | 16.7 | 90.5 | 0.0 | 0.0 | 30.0 | 12.5 | 0.0 | 23.2 |
| Original | 0.0 | 7.5 | 0.0 | 10.0 | 10.0 | 8.3 | 100.0 | 0.0 | 28.6 | 17.5 | 16.7 | 5.0 | 21.3 |

## 3.3 RESULTS ON GENERATED TESTS

Using our generation algorithms, we first construct a "Normal" subset in which each configuration closely mirrors the original FRCT questions. We then create "Easy" and "Hard" subsets by systematically adjusting parameters that modulate task difficulty. For instance, we vary the grid size for CF1, CF2, CF3, SS3, and VZ1; the noise severity for CS1, CS2, and CS3; the number of item pairs to be memorized in MA1; and the number of folds in VZ2.

We evaluate the GPT-4.1-2025-04-14 model, and the results are presented in Table 3. The model's performance increases progressively across the easy, normal, and hard subsets. Our key findings are as follows: (1) CS1–3 (object and word recognition under noise): The model achieves higher accuracy on our generated datasets compared to the original ones. We attribute this to our selection of commonly encountered objects in daily life, which likely reduces recognition difficulty. Moreover, our framework supports dynamic image updates, allowing the tests to be refreshed as needed in the future. (2) MA1 (memory test): The original version requires memorizing 21 image-number pairs, a task on which the model achieves 100% accuracy. In contrast, our hard version increases the number of pairs to 50, resulting in a substantial performance drop, highlighting the increased challenge. (3) VZ2 (paper folding test): The original dataset includes questions based on one to three folds. Our version expands this to include up to five folds, significantly increasing task complexity. The model fails to answer any of these questions correctly. These results demonstrate that our generated dataset effectively supports dynamic adjustment of test difficulty, making it suitable for evaluating increasingly capable models.

Table 4: Human performance (31 undergraduate students) on **VISFACTOR**.

| Total | CF1 | CF2 | CF3 | CS1 | CS2 | CS3 | I3 | MA1 | MV1 | MV2 | MV3 | P3 | RL2 | S1 | S2 | SS2 | SS3 | VZ1 | VZ2 | VZ3 |
|---|---|---|---|---|---|---|---|---|---|---|---|---|---|---|---|---|---|---|---|---|---|
| **78.8** | 61.7 | 56.7 | 98.3 | 55.0 | 76.7 | 75.0 | 71.7 | 100.0 | 93.3 | 93.3 | 98.3 | 91.7 | 51.7 | 83.3 | 55.0 | 98.3 | 55.0 | 96.7 | 58.3 | 63.3 | 95.0 |

## 3.4 RESULTS ON HUMAN EVALUATION

To set a baseline to interpret model performance, we conduct a human evaluation using the identical **VISFACTOR** digital protocol administered to the models. We sample 20 items per subtest, including all associated variants, yielding 1,540 questions in total. We use the same task instructions and scoring rules as for the MLLMs. We recruit 31 university students, ensuring each question is completed

by three independent participants. The resulting average human accuracy is 78.8%, with per-subtest results summarized in Table 4. This human baseline confirms a substantial performance gap between the strongest model we test, GPT-5.1, that achieves 30.17%, and university participants. Humans outperform MLLMs on nearly all subtests except RL2 (Diagramming Relationships), where success relies more on textual object knowledge, a known strength of MLLMs rather than visual reasoning.

# 4  FAILURE ANALYSIS

## 4.1  VISUAL COMPARISON OR CONCEPT RECOGNITION?

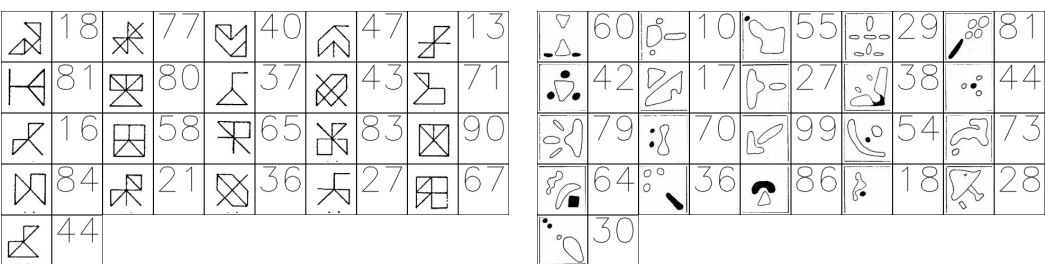

(a) MA1 generated with CF2 figures                    (b) MA1 generated with MV1 figures.

Figure 3: An example of our generated MA1 image-number pairs using CF2 and MV1 figures.

**How Do Models Master MA1?**   Given that models achieve very high accuracy on this memory test, we further investigate the mechanisms underlying their performance. An intuitive hypothesis is that models translate visual cues into high-level, human-interpretable concepts (*e.g.*, "soccer," "chair," "fish") and memorize the concept–number pairs, rather than the raw image patterns. To test this hypothesis, we use CF2-generated images, which consist only of lines arranged in a $3 \times 3$ grid, to create MA1 test cases via our automatic generation algorithm (see Fig. 3a for an example). We generate datasets with varying numbers of image–number pairs, ranging from 10 to 80, and evaluate GPT-4.1, Claude-3.7, and Qwen-VL-Max (results in Table 5). For semantically rich images, all three models maintain strong performance across different pair counts. In contrast, accuracy declines sharply with abstract CF2 images. As the number of pairs increases, GPT-4.1 demonstrates the greatest robustness, retaining 33.3% accuracy at 80 pairs. Claude-3.7 performs moderately, while Qwen-VL-Max fails at 40 pairs. We further construct test cases using abstract figures from MV1 (Fig. 3b) at 20 pairs. The three models achieve accuracies of 81.0%, 42.9%, and 54.8%, respectively, consistent with their performance on CF2-generated tests. These results suggest that models rely heavily on interpretable, concept-level representations rather than low-level visual patterns.

To ensure that the performance drop is not simply due to distributional shift, we generate extreme yet valid visual combinations using diffusion models (*e.g.*, "a horse on the moon"). In these cases, the model maintains high accuracy, further supporting our hypothesis: the model performs well as long as the visual input can be mapped to familiar, conceptual categories. This hypothesis is further supported by the analysis of the P3 Identical Pictures test, where high-performing examples typically involve easily verbalizable content, while failures are associated with visually complex and linguistically demanding patterns. These results also suggest that models struggle to interpret abstract visual patterns such as the line-based CF2 stimuli, reinforcing the idea that their success depends on concept recognition rather than low-level perception.

## 4.2  VISUAL RECOGNITION: A KEY BOTTLENECK

**Rely on Accurate Textual Descriptions.**   Our comprehensive evaluation reveals a contrast between models' strong textual reasoning capabilities and their markedly weaker visual perception performance. This disparity is exemplified by the CF3 Copying task: when models are provided with textual descriptions of line segments (starting coordinates and direction vectors), GPT-4.1 achieves perfect accuracy (100%). In contrast, performance drops sharply when the same information has to be inferred from visual inputs, with accuracy falling to just 6.2%—and no model exceeding 18.8%.

Table 5: MA1 performance of three models using different image sources and pair numbers.

| Number of Pairs | 10 | | 20 | | 40 | | 80 | |
|---|---|---|---|---|---|---|---|---|
| | MA1 | CF2 | MA1 | CF2 | MA1 | CF2 | MA1 | CF2 |
| GPT-4.1-2025-04-14 | 90.48 | 78.57 | 83.33 | 57.14 | 88.10 | 52.38 | 92.86 | 33.33 |
| Claude-3.7-Sonnet | 97.62 | 73.81 | 73.81 | 45.24 | 85.71 | 38.10 | 85.71 | 9.52 |
| Qwen-VL-Max-250408 | 97.62 | 83.33 | 88.10 | 47.62 | 90.48 | 2.38 | 73.81 | 7.14 |

**Fail to Recognize Visual Details.** In the SS2 Choosing A Path test, models consistently fail to distinguish between intersecting lines with explicit junction markers versus those without visual indicators. More critically, our generated CF3 Copying test cases (illustrated in Fig. 4) reveal that start-point identification accuracy decreases systematically with marker size variation: from 92% with large circular markers to 80% with medium markers, and ultimately 68% with small markers. This degradation suggests fundamental constraints in the models' visual attention mechanisms, where reduced visual saliency directly compromises recognition performance.

(a) Large

(b) Medium

(c) Small

Figure 4: Our generated CF3 figures with different marker sizes.

Additionally, models struggle to focus effectively on key regions, resulting in missing information. Taking a CS2 Concealed Words test of a partially erased word "women" as an example. Correct identification of the first character requires recognizing the lower left corner that differentiates "w" from "v." Similarly, identifying the fifth character as "n" relies on detecting a small vertical line in the lower right corner of the letter. Models misclassify these characters as "v" and "r," respectively, indicating its limited ability to prioritize critical local features.

**Low Sensitivity to Length, Angle, and Scale.** Models exhibit notable limitations in processing geometric shapes, particularly in assessing length, proportion, and angle. In the CF3 Copying Test, the task is to replicate lines from the left side onto a $5 \times 5$ dot matrix on the right. While models can approximate line directions, they frequently fail in perceiving their lengths. Similarly, in the VZ1 Form Board Test, although models correctly identify the need for a rectangle to construct a complex figure, they fail to select sides of the appropriate length. These results indicate that while models possess some geometric recognition abilities, they struggle with accurately gauging line lengths and proportions, limiting their performance in tasks requiring precise spatial measurements. Moreover, our analysis reveals **a bias toward diagonal orientations**: models consistently misclassified various directions as 45-degree angles. In a controlled test with 20 non-45-degree vectors (*e.g.*, vector $(2, 1)$), models achieve zero correct angular identification, consistently defaulting to the nearest 45-degree approximation. This suggests that models possess only coarse categorical representations of spatial orientation rather than continuous angular perception.

**Connection to Human Visual Cognition.** Studies in cognitive psychology have shown that more verbalization can hurt human performance in certain visual or holistic reasoning tasks (Schooler & Engstler-Schooler, 1990; Dijksterhuis, 2004; Van den Bos & Poletiek, 2008). Recent work (Liu et al., 2025a) similarly reports that CoT harms MLLM performance on tasks such as face recognition and working memory. We observe related patterns in our paper: for example, in the CF3 Copying Test, humans rely on holistic spatial judgment directly from the start to the end point ("jumping steps") rather than tracing paths sequentially, whereas MLLMs' text-based reasoning forces step-by-step traversal, leading to errors. These behaviors highlight a structural mismatch between human intuitive spatial reasoning and text-mediated reasoning in current MLLMs. For tasks with semantically meaningful objects (*e.g.*, MA1), models accurately describe visual content in their chain-of-thought, suggesting that the visual backbone captures the necessary details. However, several cognitive tasks contain spatial configurations that cannot be faithfully verbalized. For example, in VZ2 Paper Folding Test, describing the exact relative distances of holes (*e.g.*, "two holes symmetrically placed along the diagonal") is inherently insufficient, leading to incorrect predictions. Similar issues arise in CF1, CF2, and MV1, where spatial relations lack stable linguistic representations.

## 5 RELATED WORK

**Evaluation with Natural Images.**    Natural images are commonly used to evaluate the visual capabilities of MLLMs, as they more closely reflect real-world scenarios (Zhao et al., 2024; Liu et al., 2024a; Chow et al., 2025; Wadhawan et al., 2024). Recent research has emphasized MLLMs' spatial reasoning abilities (Kamath et al., 2023; Liu et al., 2023; Yang et al., 2025), including tasks such as top-view map interpretation (Li et al., 2024) and region-level depth reasoning (Cheng et al., 2024). However, we argue that natural images often introduce additional noise and variability, making them less suitable for assessing core visual competencies. While benchmarks such as Blink (Fu et al., 2024), MMT-Bench (Ying et al., 2024), HallusionBench(Guan et al., 2024), and CoreCognition (Li et al., 2025b) incorporate synthetic images for tasks like IQ testing, visual hallucination detection, and physical reasoning, their overall focus remains primarily on natural image settings.

**Evaluation with Synthetic Images.**    Synthetic images have been widely employed to evaluate the fundamental visual reasoning capabilities of MLLMs (Rahmanzadehgervi et al., 2024; Wu et al., 2024a; Chollet et al., 2025; Moskvichev et al., 2023). Prior work has leveraged tasks such as Raven's Progressive Matrices (Zhang et al., 2024; Song et al., 2024; Cao et al., 2024; Zhang et al., 2019) and the Logic Test from the Chinese Civil Service Examination (Song et al., 2025), which include puzzles conceptually related to our I3 task. VisualSphinx (Feng et al., 2025) further extends this line of work by generating puzzles structurally similar to RPMs. Mental Rotation Tests have also been frequently used (Ramakrishnan et al., 2025; Song et al., 2024), aligning with the design of our S1 and S2 tasks. In addition, synthetic images have supported evaluations of MLLMs on mathematical reasoning problems (Lu et al., 2024; Wang et al., 2025a), including polygons (Rudman et al., 2025) and graph-based challenges (Babaiee et al., 2025). Our proposed **VISFACTOR** advances this direction by providing a more comprehensive evaluation framework for core visual abilities, including 20 tests, systematically grounded in factor analysis from cognitive science. Furthermore, we implement automatic generation for 12 tests, enabling unlimited training data and ensuring the long-term scalability of the benchmark by high difficulties.

**Enhancing MLLMs' Visual Ability.**    A range of strategies have been proposed to strengthen spatial reasoning in MLLMs, including generating intermediate steps (Li et al., 2025a; Wu et al., 2024b), drawing auxiliary lines (Meng et al., 2023; Hu et al., 2024), incorporating coordinates or depth cues (Liu et al., 2025b; Cai et al., 2024), and augmenting training sets with reasoning data (Shao et al., 2024). Our approach enables automatic generation of high-quality, difficulty-controlled test cases, offering unlimited training data to enhance MLLMs' visual reasoning.

**Using Psychological Tests on AI.**    Recent studies have evaluated AI models from psychological perspectives, including behavioral analysis (Coda-Forno et al., 2024), personality (Huang et al., 2024b;a), emotion (Huang et al., 2024b), and mental disorder (Coda-Forno et al., 2023). Research has found advanced human-like abilities in AI models, including Theory-of-Mind abilities (Liu et al., 2024c; Liang et al., 2023; Huang et al., 2025) and role-playing abilities (Ng et al., 2024; Wang et al., 2024b; 2025b). Inspired from cognitive science, our work provides a comprehensive framework for evaluating foundational visual abilities.

## 6 CONCLUSION

We present **VISFACTOR**, the first factor-grounded benchmark that transposes twenty vision-centric subtests from the *Factor-Referenced Cognitive Test* battery into an automated image–text setting. A systematic evaluation of twenty MLLMs uncovers a striking gap: despite their prowess on holistic leaderboards, the best model attains only *30.17%* on **VISFACTOR**, often performing near chance on tasks that human novices solve with ease. Beyond exposing a missing substrate for genuine visual reasoning, these findings carry practical ramifications. Hallucinated perception in safety-critical applications, brittle spatial reasoning in robotics, and misaligned multimodal feedback loops all trace back to weak foundational vision. Bridging this gap will likely require *curriculum-style pre-training* that interleaves psychometric micro-tasks with natural images, *embodied or 3-D data* that grounds spatial relations, and *factor-aligned loss functions* that explicitly target low-level perceptual skills. By releasing **VISFACTOR** and its controllable-difficulty generator, we aim to catalyze these research directions and provide a rigorous yardstick for the next generation of visuocognitive AI.

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

## THE USE OF LARGE LANGUAGE MODELS

LLMs were employed in a limited capacity for writing optimization. Specifically, the authors provided their own draft text to the LLM, which in turn suggested improvements such as corrections of grammatical errors, clearer phrasing, and removal of non-academic expressions. LLMs were also used to inspire possible titles for the paper. While the system provided suggestions, the final title was decided and refined by the authors and is not directly taken from any single LLM output. In addition, LLMs were used as coding assistants during the implementation phase. They provided code completion and debugging suggestions, but all final implementations, experimental design, and validation were carried out and verified by the authors. Importantly, LLMs **WERE NOT** used for generating research ideas, designing experiments, or searching and reviewing related work. All conceptual contributions and experimental designs were fully conceived and executed by the authors.

CONTENTS

# A DATASET INFORMATION

## A.1 DATASET STATISTICS

Table 6: Basic statistics of VISFACTOR. It includes 3,046 queries covering 808 questions, which provides sufficient statistical power.

| Category | Name | ID | #Questions | #Queries | Guess Acc. |
|---|---|---|---|---|---|
| Perceptual & Closure | Hidden Figures Test | CF1 | 32 | 160 | 3.13% |
| | Hidden Patterns Test | CF2 | 80 | 400 | 3.13% |
| | Copying Test | CF3 | 64 | 64 | 4.00% |
| | Gestalt Completion Test | CS1 | 20 | 20 | 0.00% |
| | Concealed Words Test | CS2 | 50 | 50 | 0.00% |
| | Snowy Pictures | CS3 | 24 | 24 | 0.00% |
| | Identical Pictures Test | P3 | 96 | 480 | 3.13% |
| Reasoning | Figure Classification | I3 | 28 | 224 | 0.23% |
| | Diagramming Relationships | RL2 | 30 | 150 | 3.13% |
| Memory | Picture-Number Test | MA1 | 42 | 42 | 4.76% |
| | Shape Memory Test | MV1 | 32 | 128 | 6.25% |
| | Building Memory | MV2 | 24 | 120 | 3.13% |
| | Map Memory | MV3 | 24 | 96 | 6.25% |
| Visualization & Spatial Reasoning | Card Rotations Test | S1 | 20 | 160 | 0.39% |
| | Cube Comparisons Test | S2 | 42 | 168 | 6.25% |
| | Choosing A Path | SS2 | 32 | 160 | 3.13% |
| | Map Planning Test | SS3 | 40 | 80 | 1.00% |
| | Form Board Test | VZ1 | 48 | 240 | 3.13% |
| | Paper Folding Test | VZ2 | 20 | 100 | 3.13% |
| | Surface Development Test | VZ3 | 60 | 180 | 3.65% |
| | | All | 808 | 3046 | 2.89% |

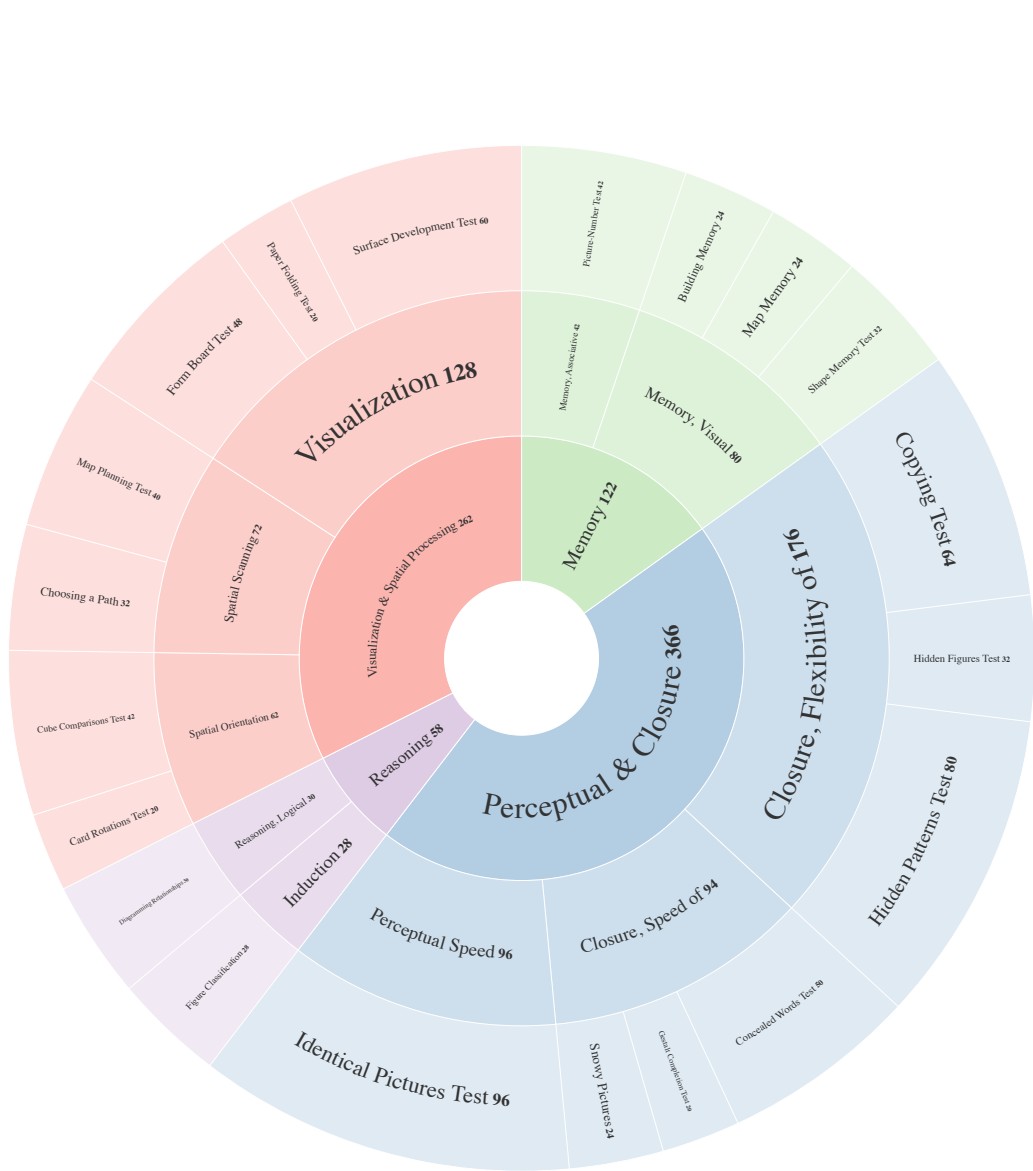

Figure 5: **VISFACTOR** integrates 20 subtests adapted from standardized human cognitive assessments. Subtests are organized into four major domains and weighted by test case count (shown numerically), which determines each segment' visual area.

## A.2 PSYCHOMETRIC VALIDITY

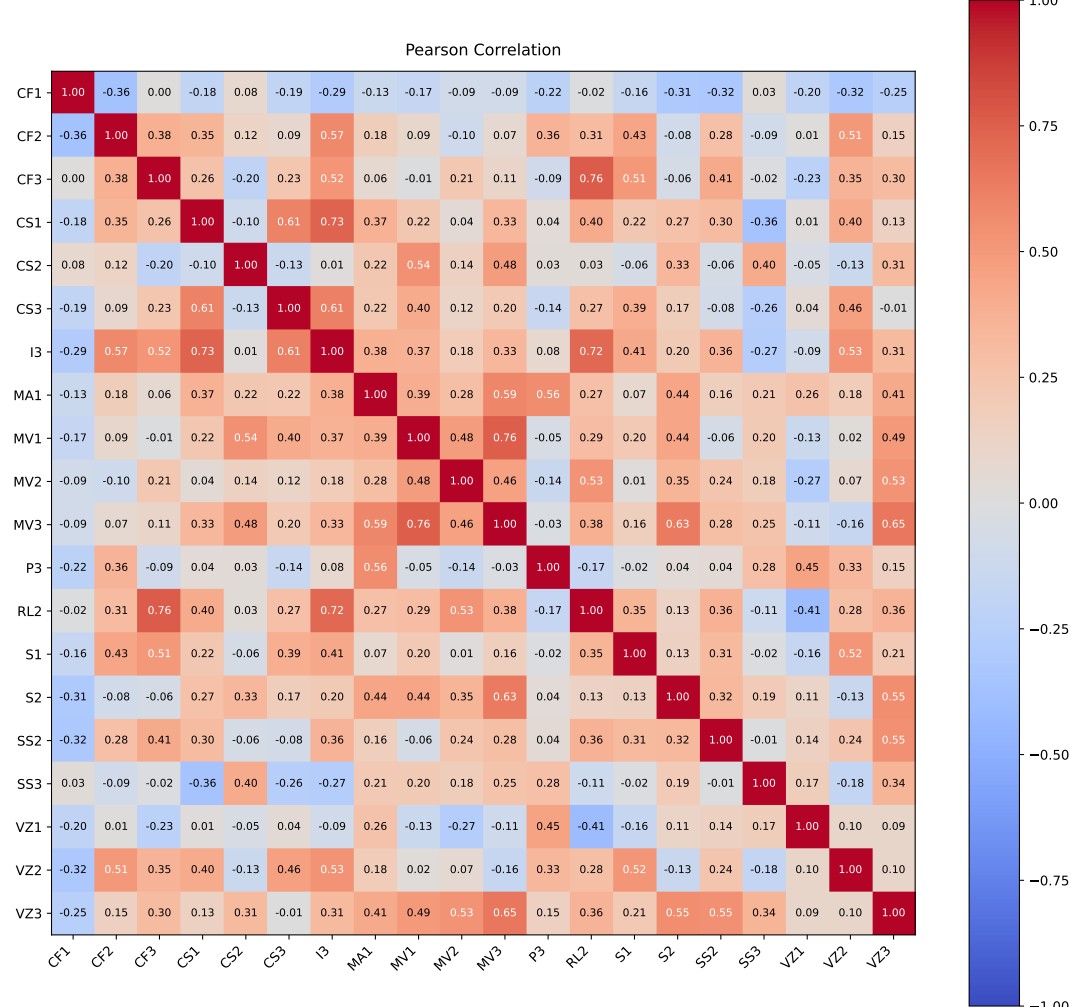

Figure 6: Pearson correlation between all subtests in **VISFACTOR**.

We provide additional psychometric analysis to assess the structure and reliability of our **VISFACTOR**. First, the overall internal consistency across the 20 subtests is ($\alpha = 0.775$), indicating that the benchmark functions as a coherent evaluation while retaining sufficient diversity across tasks. We also report model-level score homogeneity ($\alpha = 0.987$), showing that current MLLMs exhibit highly stable relative rankings across subtests; this reflects model behavior rather than a psychometric property of the test itself.

Second, the full task–task Pearson correlation matrix (shown in Fig. 6) demonstrates that no pair of subtests is excessively correlated, confirming that the benchmark is not dominated by any single dimension. For most models, pairwise correlations fall in the 0.6–0.8 range, suggesting consistent yet non-redundant task structure, whereas weaker models such as LLaMA variants produce near-zero correlations with stronger models due to uniformly low performance.

Finally, the correlation patterns also support factor isolation. Subtests associated with the same FRCT cognitive factor (*e.g.*, reasoning: I3, RL2; memory: MA1, MV1, MV2, MV3) exhibit moderately higher within-factor correlations, while cross-factor correlations remain lower. Combined with our faithful reproduction of the original item semantics and reasoning requirements, these results indicate that the digitized FRCT tasks preserve the intended cognitive factor structure.

## B  Descriptions and Prompts for all Subtests

This section introduces each subtest in detail and provides the prompts we use in **VisFactor**.

### B.1  Closure Flexibility (CF)

The Factor:

> *"The ability to hold a given visual percept or configuration in mind so as to disembed it from other well defined perceptual material."*

Flexibility of closure, a cognitive factor involving the identification of a configuration within a distracting perceptual field, has been linked to the concept of field independence, though they are not considered identical constructs. Witkin (1971) related this factor to both Thurstone's flexibility of closure (Thurstone, 1938) and Guilford's adaptive flexibility (Guilford, 1967), suggesting similarities to field independence. Royce (1973) proposed that flexibility of closure may interact with higher-order cognitive factors, while Hettema (1968) posited it as conceptually situated between flexibility and speed of closure. Wardell (1973) argued for its identity with figural adaptive flexibility. Carroll (1974) defined flexibility of closure as involving short-term memory processes that match a figure to its surrounding field, and Cattell (1971) framed it as a restructuring ability central to personality and practical intelligence.

---

**Prompt for CF1: Hidden Figures Test**

Look at the two images:

Below is the first image, one simple shape:

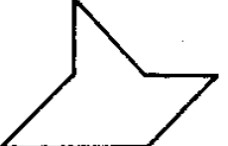

Below is the second image, a larger, complex pattern:

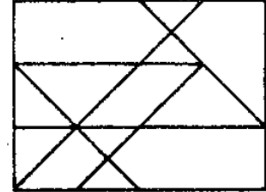

Task: Decide whether the shape in the first image is hidden anywhere inside the second image. The shape will never be rotated, flipped, or resized. The shape will always be right-side-up and exactly the same size as in the first image.

Output: Respond with only one word: "TRUE" if it is present, "FALSE" if it is not, in JSON format as follows: {"answer": YOUR_ANSWER_HERE}.

---

## Prompt for CF2: Hidden Patterns Test

Look at the two images:

Below is the first image, a model:

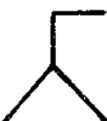

Below is the second image, a pattern:

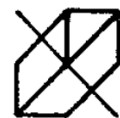

Task: Decide if the model in the first image is hidden anywhere in the pattern in the second image. The model must be in that exact position, no turning or flipping.

Output: Respond with only one word: "TRUE" if it is present, "FALSE" if it is not, in JSON format as follows: {"answer": YOUR_ANSWER_HERE}.

## Prompt for CF3: Copying Test

Look at the two images:

Below is the first image, a simple line shape:

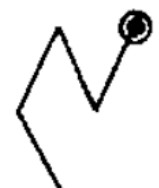

Below is the second image, a 5 times 5 grid of dots; one dot is circled as the starting point:

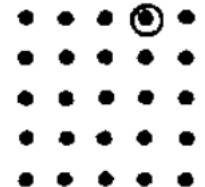

Task: Begin at the circled dot on the second image. Copy the shape shown in the first image onto the grid so that every corner of the line sits exactly on a dot. When you are done, the pattern on the grid must look the same as the shape in the first image.

Output: Respond with only a tuple, the dot you finally reach, as a (row, column) pair where the row is counted top-to-bottom and the column left-to-right, in JSON format as follows: {"answer": YOUR_ANSWER_HERE}.

## B.2 Closure Speed (CS)

The Factor:

> *"The ability to unite an apparently disparate perceptual field into a single concept."*

The concept of speed of closure refers to the ability to rapidly recognize and organize ambiguous or partially obscured visual stimuli, a process distinct from flexibility of closure, which involves identifying a known configuration within complex figures. This skill is associated with the early identification of out-of-focus and close-up images (Frederiksen, 1967), and involves long-term memory search strategies (Carroll, 1974). It has been linked to cognitive factors like restraint-timidity (Cattell, 1971) and may reflect a broader aptitude for visual scanning and cognitive-affective integration (Thurstone, 1944; Wardell, 1973; Roff, 1953; Adcock & Martin, 1971; Messick & French, 1975).

---

**Prompt for CS1: Gestalt Completion Test**

Look at the incomplete drawing below:

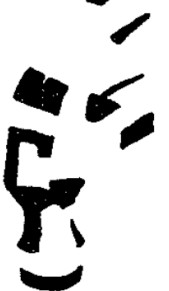

Task: Write the name of the object you think it shows.

Output: Respond with only one or two words, in JSON format as follows: {`"answer"`: YOUR_ANSWER_HERE}.

---

**Prompt for CS2: Concealed Words Test**

Look at the image below, which shows one lowercase English word, but parts of the letters are missing:

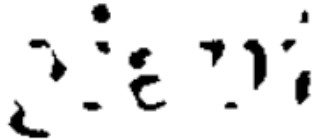

Task: Write the complete word. The word is at least four letters long. Use only lowercase letters.

Output: Respond with only the answer word, in JSON format as follows: {`"answer"`: YOUR_ANSWER_HERE}.

---

**Prompt for CS3: Snowy Pictures**

Look at this image below:

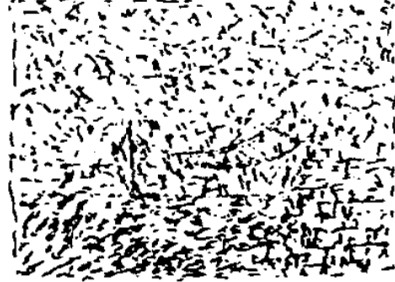

Task: Even if parts are hidden, name the main object you see.

Output: Respond with only one or two words, in JSON format as follows: {"answer": YOUR_ANSWER_HERE}.

## B.3 INDUCTION (I)

The Factor:

> *"The reasoning abilities involved in forming and trying out hypotheses that will fit a set of data."*

Research on inductive reasoning suggests it involves both concept formation and hypothesis testing, functioning as a synthesizing process (Wardell, 1973). Evidence points to several subfactors, with figure classification being particularly distinct (Harris & Harris, 1971). Guilford & Hoepfner (1966) identified 16 types of inductive ability, while Dye & Very (1968) proposed distinct inductive and symbolic-inductive reasoning factors. Though Pawlick (1966) argued that induction and general reasoning are not separate, Cattell (1971) allowed for a possible figural reasoning factor. Carroll (1974) emphasized the role of long-term memory search in induction, noting that success depends on the content of a "general logic store" and the ability to construct new hypotheses through serial operations.

---

**Prompt for I3: Figure Classification**

Look at the four images:

Below is the first image, three figures in the Group 1:

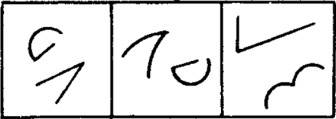

Below is the second image, three figures in the Group 2:

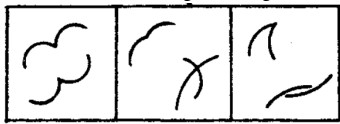

Below is the second image, three figures in the Group 3:

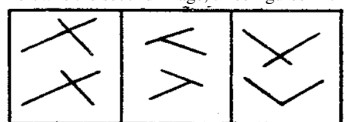

Below is the fourth image, the figure to classify:

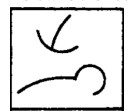

Task: Inside a group, all three figures share one rule. Different groups follow different rules. Find the rule and decide whether the figure in the fourth image belongs to Group 1, 2, or 3.

Output: Respond with only the group number (1, 2, or 3), in JSON format as follows: {`"answer"`: YOUR_ANSWER_HERE}.

---

## B.4 ASSOCIATIVE MEMORY (MA)

The Factor:

> *"The ability to recall one part of a previously learned but otherwise unrelated*
> *pair of items when the other part of the pair is presented."*

Tasks assessing this factor are similar to those used in paired-associates learning and may involve memory for non-meaningful material. This factor reflects intermediate-term memory processes, where individual differences arise from the use of strategies such as short-term rehearsal and the identification of mnemonic mediators in long-term memory (Carroll, 1974).

---

**Prompt for MA1: Picture-Number Test**

Look at the two images:

Below is the first image, the 21 picture-number pairs to memorize:

Below is the second image, a picture:

Task: Write down the number that the picture in the second image belongs to, as shown in the first image.

Output: Respond with only a number, in JSON format as follows: {"answer": YOUR_ANSWER_HERE}.

---

## B.5 VISUAL MEMORY (MV)

The Factor:

> *"The ability to remember the configuration, location, and orientation of figural material."*

Visual memory involves distinct cognitive processes beyond mere test content, as suggested by research on iconic memory, which stores visual impressions (Thurstone, 1946). While Thurstone (1946) argued that "the memorizing factor transcends the nature of the content," later studies demonstrated that visual memory is a multifaceted construct. Guilford (1967) identified six figural memory abilities, and Petrov (1970) distinguished between factors for iconic memory and short-term visual retention, indicating the presence of sub-factors within visual memory.

---

**Prompt for MV1: Shape Memory Test**

Look at the two images:

Below is the first image, memorize each shape and the way it is turned:

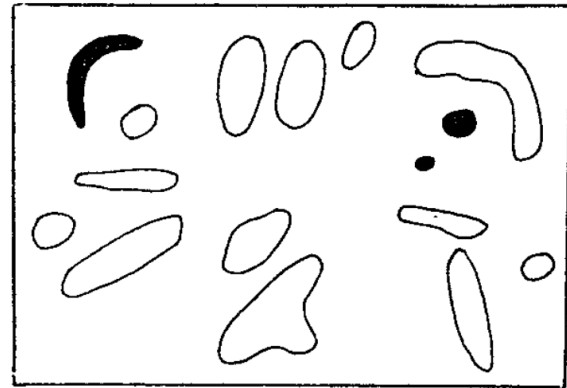

Below is the second image:

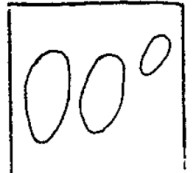

Task: Decide whether the following statement is true or false: the second image does not show any part of the first image with the same shapes in the same orientation.
(!!!) Three other prompts are: (1) the second image does not show any part of the first image with the same shapes in the same orientation (2) some part of the first image contains the second image with the same shapes in the same orientation (3) some part of the first image does not contain the second image with the same shapes in the same orientation

Output: Respond with only one word: "TRUE" or "FALSE", in JSON format as follows: {"answer": YOUR_ANSWER_HERE}.

---

## Prompt for MV2: Building Memory

Look at the two images:

Below is the first image, memorize where every building sits on this street map:

Below is the second image, the streets are the same, but each block is labeled A, B, C, D, E:

Below is the third image, a building:

Task: Decide whether the building in the third image is in block E.

Output: Respond with only one word: "TRUE" if it is, "FALSE" if it is not, in JSON format as follows: {"answer": YOUR_ANSWER_HERE}.

1566
1567
1568
1569
1570
1571
1572
1573
1574
1575
1576
1577
1578
1579
1580
1581
1582
1583
1584
1585
1586
1587
1588
1589
1590
1591
1592
1593
1594
1595
1596
1597
1598
1599
1600
1601
1602
1603
1604
1605
1606
1607
1608
1609
1610
1611
1612
1613
1614
1615
1616
1617
1618
1619

## Prompt for MV3: Map Memory

Look at the two images:

Below is the first image, memorize each map:

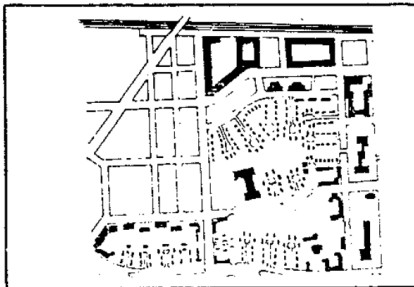 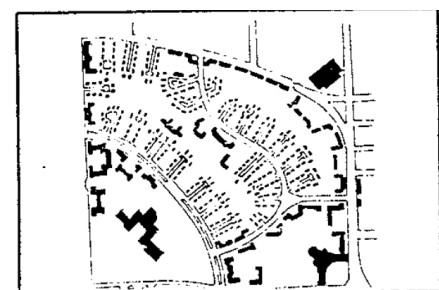

Below is the second image, a single map:

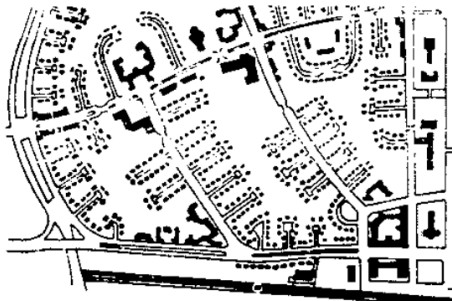

Task: Decide whether the following statement is true or false: the map in the second image appears in the first image.
(!!!) Three other prompts are: (1) the map in the second image does not appear in the first image (2) the maps in the first image contain the map in the second image (3) the maps in the first image do not contain the map in the second image

Output: Respond with only one word: "TRUE" or "FALSE", in JSON format as follows: {"answer":  YOUR_ANSWER_HERE}.

## B.6 PERCEPTUAL SPEED (P)

The Factor:

> *"Speed in comparing figures or symbols, scanning to find figures or symbols,*
> *or carrying out other very simple tasks involving visual perception."*

Perceptual speed has been described as comprising three components: (1) perceptual fluency, or the readiness with which individuals switch between alternating percepts; (2) decision speed, or the readiness of choice when the response is not fully driven by sensory input (Thurstone, 1938; Künnapas, 1969); and (3) immediate perceptual memory. Carroll (1974) defines perceptual speed as involving the temporal aspects of visual search through a field of specified elements by accessing sensory buffers. It may be related to flexibility of closure (Pawlick, 1966; Ekstrom, 1973) or to an "automatic process" factor. Additionally, (Royce, 1973) suggested it may be a subfactor of the scanning cognitive style and possibly linked to the automatization cognitive style. It may be the centroid of several subfactors (including form discrimination and symbol discrimination) which can be separated but are more usefully treated as a single concept for research purposes.

---

**Prompt for P3: Identical Pictures Test**

Look at the two images:

Below is the first image, the target object:

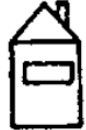

Below is the second image, the test object:

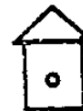

Task: Decide whether the two objects are exactly the same.

Output: Respond with only one word: "TRUE" if they are, "FALSE" if they are not, in JSON format as follows: {"answer": YOUR_ANSWER_HERE}.

---

## B.7 LOGICAL REASONING (RL)

The Factor:

> *"The ability to reason from premise to conclusion, or to evaluate the correctness of a conclusion."*

The cognitive factor historically referred to as "Deduction" (Thurstone, 1938), later termed "Syllogistic Reasoning," and also known as "Logical Evaluation", involves evaluating the correctness of presented answers rather than pure deductive reasoning (Guilford, 1967). Carroll (1974) emphasized its complexity, highlighting the need for retrieving meanings and algorithms from long-term memory and applying serial operations, with individual differences influenced by content, timing, and attentional focus on stimuli.

---

**Prompt for RL2: Diagramming Relationships**

Look at the image below:

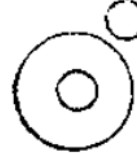

Each circle stands for one group of things. Simple rules:
1. A circle inside another: all things in the inner group belong to the outer group.
2. Circles that overlap partly: the two groups share some, but not all, things.
3. Circles that do not touch: the two groups share nothing.

Task: Decide whether the image follows these rules for the three groups: Desks, furniture, pencils.

Output: Respond with only one word: "TRUE" if it shows the relationships for the three groups, "FALSE" if it does not, in JSON format as follows: {"answer": YOUR_ANSWER_HERE}.

---

## B.8 SPATIAL RELATIONS (S)

The Factor:

> *"The ability to perceive spatial patterns or to maintain orientation with respect to objects in space."*

Research has differentiated between spatial orientation and visualization, suggesting that while spatial orientation involves perceiving figures as wholes and performing mental rotation (Zimmerman, 1954; Werdelin & Stjernberg, 1971), visualization requires more complex restructuring and serial operations (Carroll, 1974; Shepard & Metzler, 1971). Although some distinguished between spatial relations and orientation (with the latter involving the observer's body), Guilford & Hoepfner (1971) treated them as a single cognitive factor linked to egocentrism.

---

**Prompt for S1: Card Rotations Test**

Look at the two images:

Below is the first image, the target shape:

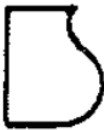

Below is the second image, the test shape:

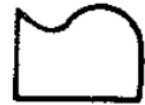

Task: The test shapes may be rotated, but they are not allowed to be flipped (mirrored). Decide whether test shape is the same shape as the target.

Output: Respond with only one word: "TRUE" if it is, "FALSE" if it is not, in JSON format as follows: {"answer": YOUR_ANSWER_HERE}.

---

1782
1783
1784
1785
1786
1787
1788
1789
1790
1791
1792
1793
1794
1795
1796
1797
1798
1799
1800
1801
1802
1803
1804
1805
1806
1807
1808
1809
1810
1811
1812
1813
1814
1815
1816
1817
1818
1819
1820
1821
1822
1823
1824
1825
1826
1827
1828
1829
1830
1831
1832
1833
1834
1835

## Prompt for S2: Cube Comparisons Test

Look at the two images:

Below is the first image, the first cube:

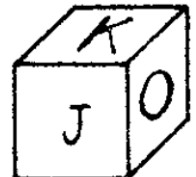

Below is the second image, the second cube:

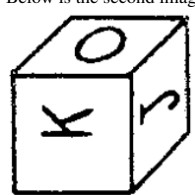

Rules:
1. Each cube has six faces. Every face shows a different letter, number, or symbol.
2. Hidden faces may show any symbols, but no symbol appears on more than one face of the same cube.

Task: Decide whether the following statement is true or false: the first cube is a certain view of the second cube after it is turned.
(!!!) Three other prompts are: (1) the first cube is not any view of the second cube no matter how it is turned (2) the second cube is a certain view of the first cube after it is turned (3) the second cube is not any view of the first cube no matter how it is turned

Output: Respond with only one word: "TRUE" or "FALSE", in JSON format as follows: {"answer": YOUR_ANSWER_HERE}.

## B.9 SPATIAL SCANNING (SS)

The Factor:

> *"Speed in exploring visually a wide or complicated spatial field."*

The ability to navigate a paper maze relies on quickly scanning for viable paths and rejecting false leads, engaging a visual search process somewhat akin to scanning text for comprehension. While sometimes associated with "planning," the process primarily reflects a willingness to visually evaluate options before committing. Carroll (1974) noted that this skill involves managing sensory input and that individuals may adopt strategies such as working backward from the goal to simplify the task.

---

**Prompt for SS2: Choosing A Path**

Look at the diagram shown in the image below:

Rules:
1. You may switch lines only where a black dot is drawn.
2. Lines that cross or touch without a dot are not connected.
3. The path must stay inside the chosen box and must not stop at a dead-end.

Task: For box E, decide if there is one continuous line that:
1. Starts at S inside that box.
2. Reaches the single circle at the top.
3. Comes back to F inside the same box without entering any other box.

Output: Respond with only one word: "TRUE" if box E meets all the rules, "FALSE" if it does not, in JSON format as follows: {"answer": YOUR_ANSWER_HERE}.

---

1890
1891
1892
1893
1894
1895
1896
1897
1898
1899
1900
1901
1902
1903
1904
1905
1906
1907
1908
1909
1910
1911
1912
1913
1914
1915
1916
1917
1918
1919
1920
1921
1922
1923
1924
1925
1926
1927
1928
1929
1930
1931
1932
1933
1934
1935
1936
1937
1938
1939
1940
1941
1942
1943

## Prompt for SS3: Map Planning Test

Look at the city map shown in the image below:

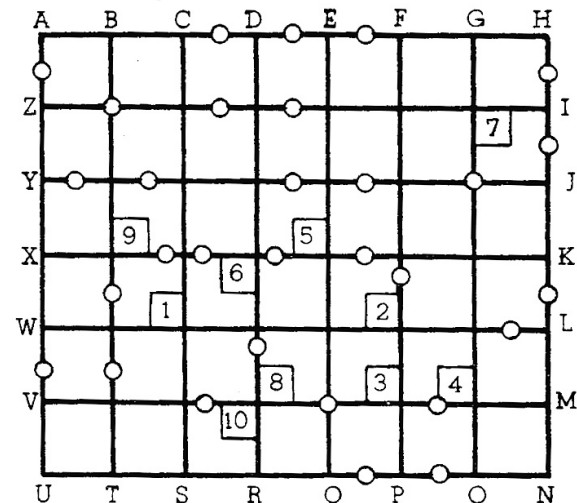

In the map:
1. Streets = black lines.
2. Circles = road-blocks (you cannot cross there).
3. Numbered squares = buildings.

Task: Find the shortest street route from F to T. Rules:
1. The route will always touch the side of one and only one numbered building.
2. Touching only a corner does not count.
3. Move only along streets (horizontal or vertical), never through circles.

Output: Respond with only one number: the number on the building your shortest route touches, in JSON format as follows:
{"answer": YOUR_ANSWER_HERE}.

## B.10 VISUALIZATION (VZ)

The Factor:

> *"The ability to manipulate or transform the image of spatial patterns into other arrangements."*

Visualization and spatial orientation are related cognitive factors, yet visualization involves mentally restructuring figures into components for manipulation, making it more complex than spatial orientation, which deals with rotating entire figures. While some researchers view visualization as a higher-order or secondary factor encompassing various spatial abilities (Cattell, 1971; Royce, 1973), others emphasize its reliance on short-term visual memory and serial processing (Carroll, 1974). Analytic strategies, such as identifying symmetry and reflection planes, are often used in visualization tasks, as illustrated by Shepard & Feng (1972)'s work on paper-folding tests.

---

**Prompt for VZ1: Form Board Test**

Look at the two images:

Below is the first image, which is the figure you must make:

Below is the second image, which are the five pieces you can use:

Rules:
1. Use 2–5 of the pieces to fill the figure exactly.
2. You may rotate pieces but do not flip them.

Task: Decide whether the Fifth piece is in the set of pieces that makes the figure.

Output: Respond with only one word: "TRUE" if it is or "FALSE" if it is not, in JSON format as follows: {"answer": YOUR_ANSWER_HERE}.

---

## Prompt for VZ2: Paper Folding Test

Look at the two images:

Below is the first image, a step-by-step drawing of a square sheet being folded (solid lines) and then punched (small circle marks):

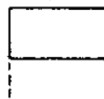 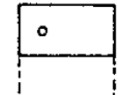

Below is the second image, the same sheet shown completely unfolded, with any holes that appear:

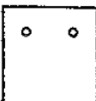

Task:
1. Mentally follow every fold in the first image exactly as drawn. Do not flip or rotate the paper except for the folds shown.
2. Imagine a hole being punched through all layers where each circle is drawn.
3. Unfold the paper, step by step, in reverse order of the folds, keeping the sheet's original orientation.
4. After it is flat, note where every hole should appear on the sheet.
5. Compare this mental result with the pattern of holes in the second image.

Output: Respond with only one word: "TRUE" if every hole (number and position) in the second image matches your mental result exactly, otherwise "FALSE", in JSON format as follows: {"answer": YOUR_ANSWER_HERE}.

## Prompt for VZ3: Surface Development Test

Look at the two images:

Below is the first image, the flat paper:

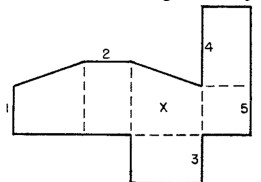

Below is the second image, the 3-D object:

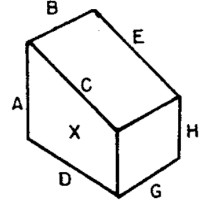

Task: Fold the flat paper in the first image on every dashed line so that the face marked X ends up on the outside of the 3-D object in the second image. Decide edge 5 on the flat paper in the first image touches which lettered edge on the 3-D object in the second image after folding.
(!!!) Decide whether the pair of one letter on the 3-D object in the second image and one number on the flat paper in the first image: (5, H) are two edges that touch each other after folding.

Output: Respond with only one letter, in JSON format as follows: {"answer": YOUR_ANSWER_HERE}.
(!!!) Output: Respond with only one word: "TRUE" if they touch, "FALSE" if they do not, in JSON format as follows: {"answer": YOUR_ANSWER_HERE}.

## C  Implementation Details

### C.1  CF1: Hidden Figures Test

We model each pattern as a graph $G = (V, E)$ embedded on an axis-aligned $m \times n$ lattice whose admissible edges join adjacent vertices (4-neighbour plus the two diagonals). Generation starts by deterministically adding the perimeter edges, thereby fixing a closed bounding rectangle and seeding a single connected component. The target edge count is then drawn from $k \sim \mathcal{N}(\mu, \sigma^2)$ with $\mu = \rho |E|$ and $\sigma = \rho_{\text{std}} |E|$ for user-specified density $\rho \in (0, 1]$ and $\rho_{\text{std}}$, and clipped to $[0, 1] \cdot |E|$. For sub-pattern detection we represent the user-supplied "model" as its own edge set and enumerate all translations obtained by aligning any model vertex with any pattern vertex; containment reduces to a constant-time subset test per translation, which is tractable for the small grids used here and yields exact, translation-invariant matches without recursion or graph isomorphism search.

### C.2  CF2: Hidden Patterns Test

We introduce a graph-based generator that operates on an $m \times n$ lattice. We first enumerate the complete set $\mathcal{E}$ of admissible edges—unit horizontal, vertical, and diagonal connections between adjacent lattice nodes—yielding $E = |\mathcal{E}|$ potential segments. To guarantee global connectivity, we draw a uniformly random spanning tree $T \subset \mathcal{E}$ by performing a depth-first search with randomized successor order; this yields exactly $N - 1$ edges, where $N = mn$ is the number of nodes. Desired edge density is controlled by sampling a target count $k \sim \mathcal{N}(\mu, \sigma^2)$ with $\mu = \rho E$ and $\sigma = \rho_{\text{std}} E$ for user-specified density $\rho \in (0, 1]$ and $\rho_{\text{std}}$; the sample is clipped to $[N - 1, E]$. We then augment $T$ with $k - (N - 1)$ additional edges drawn without replacement from $\mathcal{E} \setminus T$, producing a connected graph $G = (V, E_G)$ whose expected density equals $\rho$.

### C.3  CF3: Copying Test

We develop a procedural grid-walk generator that produces paired images. Each instance begins by laying out an $m \times n$ lattice whose node coordinates are computed analytically from a single size parameter, ensuring scale-invariance across resolutions. A start node is selected uniformly at random and a self-avoiding walk is grown whose length is drawn from a user-specified interval $[\text{min\_steps}, \text{max\_steps}]$. At every extension step, the candidate set comprises all yet-unvisited lattice nodes; candidates that would yield a line segment collinear with any existing segment in the path are deterministically excluded via a zero-cross-product test, preventing visual overlap and ensuring topological diversity. Two images are rendered, a reference grid with the start node circled, and a path image of identical dimensions that shows only the start node and the resulting non-collinear walk.

### C.4  CS1: Gestalt Completion Test

We begin by curating object silhouettes and their labels from public image repositories. Each image is partially occluded with randomly oriented white strokes whose number and width scale linearly with a severity coefficient $s \in [0, 1]$.

### C.5  CS2: Concealed Words Test

We synthesize a tunable corpus of occluded word images by sampling from the `top_n_list` in the `wordfreq` Python library, retaining alphabetic tokens whose lengths fall within a user-defined interval and converting them to lower-case. Each word is rendered on a white canvas and then obfuscated by superimposing straight white line segments and circular blotches drawn at random positions. The number, thickness, and radius of these artifacts increase linearly with a continuous severity parameter $s \in [0, 1]$, providing precise control over the level of visual concealment.

### C.6  CS3: Snowy Pictures

Building on the silhouettes and labels introduced in CS1, we corrupt every input image in two successive steps. First, we overlay $n_r$ white rectangles whose side lengths are sampled uniformly up

to a fixed fraction of the image's shorter edge, disrupting local continuity. Next, we draw $n_\ell$ short, randomly oriented black line segments that imitate dense, edge-like clutter. Both $n_r$ and $n_\ell$ scale linearly with a severity parameter $s \in [0, 1]$.

### C.7   MA1: PICTURE-NUMBER TEST

Also building on the source from CS1, we first draw $N$ unique items without replacement and an equal-sized set of distinct two-digit integers $\{10, \dots, 99\}$. The two cells are concatenated horizontally to form an atomic pair, and all pairs are then tiled row-major into an $r \times c$ grid with $rc \geq N$ and $|r - c|$ minimized to approximate isotropy, yielding a visually balanced layout regardless of $N$. A uniformly random pair is sampled to provide a query image and its label, while the full canvas supplies rich contextual clutter.

### C.8   S1: CARD ROTATIONS TEST

We devise a lightweight generator that first samples a simple, non-self-intersecting polygon by drawing *i.i.d.* polar radii and sorted angles, and repeatedly rejecting candidates whose (i) shortest edge falls below a minimum-length threshold and (ii) consecutive edge-length differences are within a tight tolerance—two filters that jointly suppress near-symmetries and visually imperceptible edges. We optionally apply a horizontal mirror, then rotate it by a uniformly random angle before centrally cropping back to the original spatial extent. From every base polygon we generate $N$ views and record a binary label indicating whether the transformation involved only rotation (true) or a mirror-plus-rotation (false).

### C.9   S2: CUBE COMPARISONS TEST

To decide whether two partial observations correspond to the same physical cube, we cast the problem as a constrained search over the 24 right-handed orientations of a cube in $\mathbb{Z}^3$. We first "pin" the first view as the reference orientation—its Up, Front and Right faces become the intrinsic Up, Front, Right faces of the cube—which lets us record its three symbols and their rotations in a baseline face–rotation table. For each of the 24 global orientations we then (i) map the observer's local axes to intrinsic cube faces via simple cross-product geometry, (ii) transform the second view's reported rotations into the reference frame by adding a pre-computed 90° offset that aligns local "Up" vectors, and (iii) enforce two consistency constraints: (a) the same intrinsic face observed twice must carry identical symbols whose rotations are equivalent under the symbol's symmetry class (4-fold, 2-fold, or asymmetric), and (b) a symbol may not appear on two different faces. Finally, we randomly generate such three-face views and render them as perspective-correct 3-D cube images.

### C.10   SS3: MAP PLANNING TEST

We model the city layout as a rectangular $m \times n$ lattice in an undirected graph, where each vertex represents a street intersection and each edge a unit-length street segment. From the fully connected lattice we remove a user-specified fraction $r$ of edges, chosen uniformly at random, and tag their mid-points as circular "road-blocks," thereby enforcing non-traversable segments while preserving the geometry for visualization. $N_B$ quarter-square buildings are sampled without replacement from the $(m - 1)(n - 1)$ grid cells, along with the two edges each of them touches. Perimeter intersections are labeled in clockwise order using spreadsheet-style indices (A–Z, AA, AB, . . . ), after which start–end terminals are selected by random permutation until exactly one shortest path exists between them, which guarantees uniqueness while avoiding exhaustive search. The final instance thus comprises a sparse planar graph with a provably unique geodesic, alongside metadata for blocked edges, buildings and perimeter labels.

### C.11   VZ1: FORM BOARD TEST

We design an automatic pipeline that transforms an arbitrary lattice-defined polygon into a "dissect-and-assemble" puzzle while guaranteeing a unique solution under rotation and translation. The target shape is first specified on an $n \times n$ integer grid as an ordered list of boundary edges. A random integer $k \in \{3, 4, 5\}$ determines the number of genuine solution pieces. Starting from the full

polygon, we iteratively bisect the currently largest fragment with straight grid-aligned cuts whose slopes are limited to $+\infty, 0, \pm 1, \pm 2, \pm 3$. Each cut is accepted only if it produces two valid polygons, and the process terminates as soon as $k$ fragments are obtained. To generate the remaining $5 - k$ distractor pieces, we re-cut one randomly chosen solution fragment, rejecting candidate fragments whose areas coincide with any existing piece, thereby ensuring that no spurious subset of distractors can reconstruct the target.

## C.12 VZ2: PAPER FOLDING TEST

Starting from a unit-square sheet discretized into an $n \times n$ grid, our algorithm iteratively selects a random fold axis—horizontal, vertical, or an arbitrary offset diagonal of the form $y = \pm x + c$. At each step, the square is partitioned by this axis; the half-plane judged closest to the sheet's geometric center remains stationary, while the opposite half is reflected via an analytic mapping that preserves affine structure. Crucially, we maintain (i) a "Polygon" describing the current outer outline, (ii) an ordered list of internal edges and crease lines, and (iii) the exact set of point holes. These entities are updated by reflecting only those primitives that lie on the moving half and clipping fold-axis segments to the unfolded outline, guaranteeing topological correctness even for degenerate or off-center folds. The complete state history enables deterministic reverse unfolding to generate the answer: holes are "back-propagated" by conditional reflection.

# D  SOLUTIONS FROM GPT-4O

### Solution to CF1: Hidden Figures Test

1. **Identify the Pattern**: Examine the small shape in the first image and record its exact pixel or cell configuration (e.g., a 2D grid of colors or pixels).
2. **Scan the Larger Image**: Systematically slide a window of the same size as the first image over the second image, checking each possible sub-region.
3. **Compare**: For each sub-region, check if it matches the pattern from the first image exactly—no rotation, flip, or size change allowed.
4. **Decision**: If an exact match is found, output '"answer": "TRUE"'. If no match is found after scanning the entire larger image, output '"answer": "FALSE"'.

### Solution to CF2: Hidden Patterns Test

1. **Identify Model Dimensions**: Note the size (rows x columns) of the model in the first image.
2. **Scan Pattern Image**: Slide a window of the same dimensions across the second image (top-left to bottom-right).
3. **Check for Exact Match**: At each position, compare the sub-section of the pattern with the model.
4. **No Rotation or Flip**: Ensure the comparison uses the model as-is, without any transformations.
5. **Return Result**: If an exact match is found, return '"answer": "TRUE"'. Otherwise, return '"answer": "FALSE"'.

### Solution to CF3: Copying Test

1. **Observe the shape** in the first image and break it into straight line segments along the grid.
2. **Start at the circled dot** in the second image.
3. **Trace the same movements** (up/down/left/right/diagonal) from the start point, replicating the shape exactly by placing corners on the grid dots.
4. **Count steps carefully** to ensure each corner aligns with a grid dot as in the original shape.
5. **Record the final dot** reached after completing the entire shape.

## Solution to CS1: Gestalt Completion Test

1. **Observe the drawing**: Look closely at the curved and linear segments to infer what object is being sketched.
2. **Look for familiar outlines**: Identify key features—shapes, proportions, and positioning—that suggest a common object (e.g., wheels, body, handles).
3. **Mentally complete the figure**: Use the partial lines to visualize what the full object would look like.
4. **Identify the object**: Based on the partial sketch, determine the most likely object.

## Solution to CS2: Concealed Words Test

1. **Analyze the visible fragments**: Identify parts of letters that are still visible and match them to possible lowercase letters.
2. **Visualize missing parts**: Mentally fill in the gaps based on typical letter structures.
3. **Look for patterns**: Combine identified letters into a coherent word, considering common English words.
4. **Verify length**: Ensure the word is at least four letters long and uses only lowercase letters.

## Solution to CS3: Snowy Pictures

1. **Identify visible features**: Focus on the parts that are not hidden—shape, color, structure, or details that hint at the object.
2. **Infer the whole object**: Use context and symmetry to mentally complete the object, even if part is obscured.
3. **Choose the most likely object**: Based on the visible portion and common objects with that appearance.

## Solution to I3: Figure Classification

1. **Examine Group 1 and Group 2 figures**: Look for common traits shared within each group (e.g., shape count, orientation, fill patterns, symmetry).
2. **Identify the rule per group**: Determine what consistent rule applies to all three figures in each group (e.g., all shapes have a diagonal line, or all contain a specific number of elements).
3. **Compare rules between groups**: Make sure the rule is not shared across groups—each group must have a distinct rule.
4. **Analyze the figure to classify**: Determine which group's rule the new figure follows.
5. **Assign it to the correct group**: Match the figure to the group with the corresponding visual rule.

## Solution to MA1: Picture-Number Test

1. **Study the 21 picture-number pairs** in the first image: Memorize or note the associations between each unique picture and its corresponding number.
2. **Examine the picture in the second image**: Identify the object or scene shown.
3. **Match the second image** to one of the 21 pictures from the first image by comparing visual features.
4. **Retrieve the associated number** from the first image that corresponds to the matched picture.
5. **Return the number** in the required JSON format.

## Solution to MV1: Shape Memory Test

1. **Memorize the shapes and orientations** in the first image: Focus on each shape's design and the direction it's facing (rotation or reflection).
2. **Examine the second image**: Identify the specific shape and its orientation shown here.
3. **Compare it to the memorized shapes**: Look for an exact match in both shape and orientation from the first image.
4. **Evaluate the statement**: Determine if the given claim (e.g., "This shape matches one from the first image") accurately reflects what is shown.
5. **Decide if the statement is TRUE or FALSE** based on your comparison.

## Solution to MV2: Building Memory

1. **Memorize the street map** in the first image: Note the position of each unique building relative to the street layout.
2. **Study the block labels** in the second image: Understand how the blocks (A–E) correspond to the same street layout from the first image.
3. **Identify the building** in the third image: Match its shape, size, and features to one on the original street map.
4. **Locate that building** on the labeled block map from the second image.
5. **Determine if it is in the specified block**: Compare its actual position to the named block in the question.

## Solution to MV3: Map Memory

1. **Memorize the maps** in the first image: Focus on the layout of walls, paths, and any unique structures in each map.
2. **Examine the single map** in the second image: Pay attention to the same features—structure, layout, and orientation.
3. **Compare the second map** to the ones memorized: Check for exact matches or close similarities, including possible rotations or reflections.
4. **Evaluate the statement**: Determine whether it correctly asserts a match (or lack thereof) between the second map and any from the first image.
5. **Answer TRUE or FALSE** depending on whether the claim aligns with your comparison.

## Solution to P3: Identical Pictures Test

1. **Study the target object** in the first image: Note its overall shape, orientation, components, and details.
2. **Examine the test object** in the second image: Observe the same features—shape, structure, and orientation.
3. **Compare both objects** precisely: Check for any differences in angles, positioning, parts, or missing elements.
4. **Determine exact match**: Decide if the test object is an identical copy of the target object in all aspects.

## Solution to RL2: Diagramming Relationships

1. **Understand the group relationships described** in the statement (e.g., one group is a subset of another, or groups partially overlap or are completely separate).
2. **Interpret the image**: Analyze how the circles are arranged—nested, overlapping, or separate.
3. **Match visual arrangement to rules**:
- Inner circle → subset.
- Partial overlap → some shared elements.
- No contact → no shared elements.
4. **Compare stated relationships to the diagram**: Check if the image visually represents the described group relationships.

## Solution to S1: Card Rotations Test

1. **Analyze the target shape**: Note the structure, angles, and direction of each segment.
2. **Examine the test shape**: Look closely at its geometry and orientation.
3. **Mentally rotate the test shape**: Try 90°, 180°, and 270° rotations to see if it matches the target shape.
4. **Avoid flipping**: Ensure no mirroring is involved—only rotation is allowed.
5. **Compare for an exact match**: If any rotation results in a match with the target shape, it's the same.

## Solution to S2: Cube Comparisons Test

1. **Observe the visible faces** on both cubes: Note which letters, numbers, or symbols are shown.
2. **Apply the rule**: Each symbol can appear only once per cube.
3. **Check the statement**: Determine if it claims, for example, that both cubes can show the same symbol on different faces.
4. **Assess the possibility**: If a symbol appears on both cubes, that's allowed. But if the statement implies a repeat **on the same cube**, and that symbol is already visible, then it's **FALSE**.
5. **Use logic for hidden faces**: Hidden symbols must be unique on the same cube, so validate the claim accordingly.

## Solution to SS2: Choosing A Path

1. **Identify box [X]**: Locate the specific box labeled with the given identifier (e.g., A, B, etc.).
2. **Trace the path from S**: Begin at the "S" (Start) within that box and follow valid line connections.
3. **Follow only allowed connections**: Switch lines only at black dots; ignore line crossings without dots.
4. **Reach the single circle**: Confirm there's a path to the circle at the top within the same box.
5. **Return to F**: Ensure a path continues from the circle back to "F" (Finish) without dead ends or exiting the box.
6. **Check for continuity**: Confirm the entire path is unbroken, valid, and obeys all rules.

## Solution to SS3: Map Planning Test

1. **Identify the starting and ending points**: Determine the origin location mentioned (e.g., an address or landmark) from the input.
2. **Trace valid street paths**: Follow black lines only, avoiding any paths blocked by circles (road-blocks).
3. **Evaluate shortest routes**: Try different paths from start to end, counting steps while obeying street and block rules.
4. **Check building contact**: Ensure each route touches the side (not just corner) of only **one** numbered building.
5. **Choose the shortest valid path**: Among all such routes, pick the one with the fewest steps and identify the building it touches.
6. **Output the building number**: Return only the number of the building that the shortest valid path touches.

**Solution to VZ1: Form Board Test**

1. **Examine the target figure and puzzle pieces**: Analyze the target shape's structure and the five given pieces, noting their shapes and how they might fit together.
2. **Identify usable pieces**: Consider each piece's potential rotations (no flipping), and determine which combinations (using 2–5 pieces) can fill the target figure exactly.
3. **Test combinations logically**: Mentally or visually try valid combinations of pieces that can recreate the target shape without gaps or overlaps.
4. **Check for inclusion of the highlighted piece**: For each valid solution, confirm whether the highlighted ('¡ADDITIONAL_0¿') piece is included.
5. **Decide TRUE or FALSE**: If the highlighted piece appears in at least one valid combination that forms the figure, answer TRUE; otherwise, answer FALSE.

**Solution to VZ2: Paper Folding Test**

1. **Analyze the folding sequence**: Follow each fold step-by-step in the order shown, ensuring the paper orientation is preserved.
2. **Track fold layers**: At each fold, visualize how layers stack on top of each other.
3. **Simulate hole punches**: When a hole is punched through all layers, determine where it would pass through based on the stacked layers.
4. **Unfold in reverse order**: Carefully reverse each fold, mentally tracking where the hole marks should appear on the unfolded paper.
5. **Compare with the second image**: Check that the number, position, and symmetry of all holes match exactly with your mental unfolding.

**Solution to VZ3: Surface Development Test**

1. **Identify face X** on the flat paper: This face must end up on the outside of the folded 3D object.
2. **Visualize folding**: Mentally fold the paper along the dashed lines to form the 3D shape shown in the second image, ensuring X is on the correct face.
3. **Track edge positions**: Focus on the specified edge (e.g., "edge A") and determine how it moves and which edge it connects to after folding.
4. **Compare with the 3D object**: Match the final position of the folded edge to the labeled edges on the 3D object.
5. **Identify the corresponding edge**: Find the lettered edge that directly touches the given flat paper edge after all folds are complete.

## E  FAILURE CASES

**Failure of CF3: Copying Test (Length)**

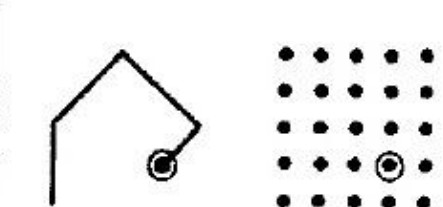

Step-by-Step Solution:
1. I'm starting at the dot at (4, 4). The first line goes diagonally up and to the right, so I take one step to (3, 5).
2. Then, there's a longer line going back up towards the left. I follow that diagonal path all the way up to (1, 3).
3. From there, the shape turns down and to the left, so I make a small diagonal move to (2, 2).
4. Finally, the last part is just a straight vertical line going down, which brings me to (3, 2).
The final coordinate is (3, 2).

**Failure of CF3: Copying Test (Angle)**

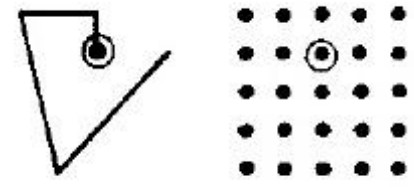

Step-by-Step Solution:
1. Alright, starting at the circled dot which is at (2, 3). First, I see a short vertical line going straight up, so I'll move one spot to (1, 3).
2. Then the line turns sharply to the left, going straight across to (1, 1).
3. From there, it drops down in a long diagonal. It looks like a typical 45-degree angle, so if I follow that path down and to the right, I go through (2, 2) and (3, 3) and land on (4, 4).
4. Finally, there's a little tail pointing up and right, so I take one diagonal step to finish at (3, 5).
The final coordinate is "''answer': (3, 5)"').

## Failure of CF3: Copying Test (Cannot Jump Steps)

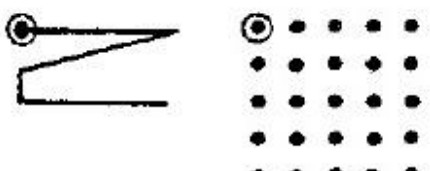

Step-by-Step Solution:
1. **Identify Start:** I locate the starting point on the grid at circle **(1, 1)**.
2. **First Segment:** I look at the first long line in the image. It looks like a diagonal line pointing down and to the right (approximately 45 degrees). I will follow this direction, moving from (1, 1) to **(3, 3)**.
3. **Second Segment:** From (3, 3), I see a short line going straight down. I move one step down to **(4, 3)**.
4. **Third Segment:** Finally, I see a small line going to the right. I move one step right to **(4, 4)**.
5. **Conclusion:** The final position is **(4, 4)**.

## Failure of VZ2: Paper Folding Test (Insufficient Textural Descriptions)

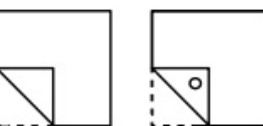 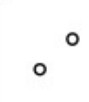

Step-by-Step Solution:
1. **Initial Setup**:
- The square sheet starts flat.
- The first fold is made by folding the bottom-left corner up to the top-right corner, creating a triangular fold.

2. **Punching the Hole**:
- After the fold, a hole is punched through all layers at the marked position (near the center of the folded triangle).

3. **Unfolding the Sheet**:
- Reverse the fold to return the sheet to its original flat state.
- Since the hole was punched through all layers of the folded sheet, it will create multiple holes when unfolded.

4. **Analyzing the Hole Placement**:
- The fold creates two layers of paper.
- A single punch through the folded sheet results in two holes when unfolded.
- The two holes will be symmetrically placed along the diagonal fold line.

5. **Comparing with the Second Image**:
- The second image shows two holes symmetrically placed along the diagonal of the square sheet.
- The positions of the holes in the second image match the expected result from the mental process.

Final Answer:
The second image matches the mental result exactly. "answer": "TRUE"

## F  COPYRIGHT ISSUES

According to the licensing information provided by ETS, the PDF version of the Kit of Factor-Referenced Cognitive Tests is made available to researchers for academic use. The licensing agreement specifies royalty requirements for researchers who reproduce physical copies of the specimen test booklets for test administration. Our work does not involve administering the tests to human participants, printing, or reproducing test forms. We only use AI to process the visual contents of the scanned items directly from the publicly accessible PDF.

However, we fully acknowledge that the availability of the PDF does not imply that the underlying FRCT items are in the public domain, nor that we are permitted to redistribute copyrighted test content. Therefore, we do not release any original FRCT items. To ensure full compliance, our public release includes only derived and non-copyrighted materials, including our generated items, metadata, and model outputs or analyses. This approach allows our work to remain entirely within the bounds of permissible academic use while avoiding any redistribution of proprietary test materials.

## G  LIMITATIONS

**Psychometric Purity.**  VISFACTOR inherits the FRCT assumption that each sub-test isolates a single latent visual factor (Ekstrom et al., 1974; 1975). In reality, human cognition is highly inter-dependent: even a seemingly "pure" mental-rotation item also taps working memory, executive control, and verbal encoding. Sub-test scores should therefore be read as *upper-bound indicators* of factor competence, not as proofs of modularity. For the same reason, factor-level comparisons across models must be interpreted with caution, especially when subtle prompt differences can shift the mixture of underlying skills that a model exploits.

**Digitization Gap.**  The original FRCT was administered on paper, under timed and proctored conditions. Our pipeline converts items to separate images and accepts typed responses, eliminating motor demands but also removing contextual cues such as page layout and time pressure. We also simplify the original instructions for MLLMs. These changes inevitably alter item difficulty, so direct numerical comparisons with legacy human norms are inappropriate.

**Missing Contemporary Human Baseline.**  We have not re-collected human performance under the digital protocol, leaving open questions about the relative difficulty of the adapted items and about ceiling effects that may mask model progress. Gathering calibrated human baselines—ideally across age groups and devices—would help normalize model scores and identify items whose difficulty distribution shifted during digitization.

