# OpenReview forum: "Castle-in-the-Air: Evaluating MLLM Visual Abilities on Human Cognitive Benchmarks"
_ICLR.cc/2026/Conference — Submitted to ICLR 2026_

### Official Review · Reviewer_giwN · 2025-10-27

**Soundness:** 2
**Presentation:** 2
**Contribution:** 2
**Rating:** 4
**Confidence:** 4

**Summary:**

This paper introduces VISFACTOR, a benchmark that digitizes 20 vision-centric subtests from FRCT, a well-established cognitive psychology assessment. It covers four domains of human visual cognition: Visualization and Spatial Processing, Perceptual and Closure, Memory, and Reasoning. Additionally, it uses parametric generation to automatically construct unlimited, difficulty-controllable test cases for applicable subtests. Furthermore, evaluations of 20 frontier MLLMs based on VISFACTOR show that the best-performing model only achieves a score of 25.19%.

**Strengths:**

1. This paper presents the benchmark that grounds MLLM assessment directly in human cognitive factors, thereby infusing psychometric rigor into multimodal evaluation.
2. The paper digitizes all FRCT visual items, designs targeted item variants to avoid random guessing biases, and introduces controllable-difficulty item synthesis specifically for the most challenging subtests—addressing the limitation of finite original test items and enabling scalable, gradient evaluation.
3. Additionally, the paper reduces the average random guessing accuracy via targeted format optimizations (e.g., decomposed multiple choice, grouped consistency, and other rule-based strategies), effectively minimizing score inflation from luck and enhancing the reliability of evaluation results.

**Weaknesses:**

1. While a parametric generator and "controllable difficulty" design are introduced, the work lacks explicit verification that these difficulty gradients align with human cognitive standards—undermining the benchmark’s validity for mapping model performance to human-like visual reasoning.
2. Though formats like "decomposed multiple choice" reduce random guessing, they raise demands on models’ language logic. Subtle wording differences may cause errors from language misunderstanding (not poor visual reasoning), distorting assessments of true visual capabilities.
3. The dataset  lacks detailed specs for each type’s sample size and clear evaluation criteria—this ambiguity hurts result reproducibility and makes assessing sample statistical sufficiency difficult.

**Questions:**

1. The paper states that VISFACTOR is derived from FRCT, a well-established cognitive psychology assessment. Does it provide data or analysis to confirm that the performance of human participants on VISFACTOR is consistent with their performance on the original FRCT?
2. The paper uses parametric generation to create unlimited, difficulty-controllable test cases. What methods or experiments were conducted to verify that these automatically generated cases can accurately distinguish differences in visual cognitive capabilities (e.g., between different MLLMs or between MLLMs and humans)?

---

> ### Author Response · Authors · 2025-11-20
> **Official Response to Reviewer giwN (1)**
>
> We sincerely thank the reviewer for the positive assessment. We appreciate your recognition of our cognitive-factor grounding, our careful digitization and extension of FRCT items, and our efforts to reduce random-guessing biases. We address your specific concerns below.
>
> # [W1] Are our generated tests more difficult?
> > While a parametric generator and "controllable difficulty" design are introduced, the work lacks explicit verification that these difficulty gradients align with human cognitive standards—undermining the benchmark’s validity for mapping model performance to human-like visual reasoning.
>
> Thank you for raising this important point regarding the validity of our difficulty gradients. Our benchmark’s difficulty settings are directly aligned with well-established human cognitive principles used in classical psychometric tests. Specifically:
> - VZ2 (Paper Folding Test): Increasing the number of folding steps is known to raise cognitive load by requiring more mental transformations and longer reasoning chains.
> - CF1–3 / SS3 (Grid-based tasks): Larger grid sizes inherently contain more visual details and information units, increasing perceptual and working-memory demands.
> - CS1–3 (Noise manipulation): Higher noise levels make target recognition more challenging, a widely validated mechanism in visual perception research.
> - MA1 (Memory tasks): Increasing the amount of content that must be retained naturally elevates memory load and task difficulty.
>
> These manipulations mirror the difficulty-control mechanisms used in established human cognitive assessments, ensuring that our benchmark's difficulty gradients are consistent with human perceptual and reasoning complexities. We also do a small human study verifying the harder questions do take a longer time for humans to solve (please see the next response to Q2 below).
>
> # [Q2] Differences in visual cognitive capabilities.
> > The paper uses parametric generation to create unlimited, difficulty-controllable test cases. What methods or experiments were conducted to verify that these automatically generated cases can accurately distinguish differences in visual cognitive capabilities (e.g., between different MLLMs or between MLLMs and humans)?
>
> Thank you for raising this important point. We provide additional analyses below:
>
> ### 1. Human study on the time taken to solve questions.
>
> To validate that our parametric generation produces difficulty levels that meaningfully reflect visual–cognitive effort, we conduct a small human study with three participants using Paper Folding Tests. **Humans require 7.44s on average to solve easy cases (one fold) and 25.59s for hard cases (five folds),** showing a clear and consistent separation in perceived difficulty.
>
> ### 2. Internal consistency of the benchmark.
>
> We computed Cronbach’s α across the 20 FRCT-derived subtests:
> - **Benchmark internal consistency: α = 0.78,** indicating that the test functions as a coherent assessment while preserving the multidimensional structure of FRCT.
> - **Model-level score homogeneity: α = 0.99,** showing that relative model rankings are stable across subtests. We report this only as an observation of current model behavior rather than a psychometric property of the test.
>
> ### 3. Inter-item and inter-task correlations.
>
> Full Pearson correlation matrices (Appendix, page 19) show that no pair of subtests exhibits excessively high correlation, confirming that the benchmark is not dominated by a single visual skill. Correlations across models fall in the 0.6--0.8 range, indicating consistent yet non-redundant task structure; LLaMA-based models show near-zero correlations with stronger models due to uniformly low performance. Tasks mapped to the same FRCT factor (e.g., reasoning: I3, RL2; memory: MA1, MV1–3) exhibit higher within-factor correlations and lower cross-factor correlations, consistent with the intended factor structure. This supports that the digitized and parametrically varied items continue to isolate the relevant cognitive abilities.
>
> # [W2] Wording of task descriptions.
> > Though formats like "decomposed multiple choice" reduce random guessing, they raise demands on models’ language logic. Subtle wording differences may cause errors from language misunderstanding (not poor visual reasoning), distorting assessments of true visual capabilities.

---

> > ### Author Response · Authors · 2025-11-20
> > **Official Response to Reviewer giwN (2)**
> >
> > We appreciate the reviewer’s concern regarding the potential influence of linguistic complexity on model performance. As we stated in Section 2.2, to minimize such confounding factors, we took deliberate steps to simplify and standardize all instructions. Specifically, we used LLM-rewritten prompts and **iterated on them two times to ensure clarity and reduce unnecessary verbosity.**
> >
> > Regarding the point about “decomposed multiple-choice” formats: converting MCQs into yes/no questions **reduces the linguistic load rather than increasing it.** MCQs require parsing 4–5 candidate options, whereas yes/no formats require understanding only a single statement. Thus, the reformulation makes the task linguistically simpler and less susceptible to errors arising purely from language comprehension.
> >
> > To further isolate whether errors stem from linguistic misunderstanding rather than visual reasoning, we conducted an ablation study: we gave GPT-4o only the task descriptions without images and asked it to provide solutions. **Across all 20 tasks, the produced solutions were coherent and logically correct.** This indicates that the model understands the task semantics and linguistic instructions. **The full outputs are included in Appendix Section D.**
> >
> > # [W3] Dataset statistics and evaluation setups.
> > > The dataset lacks detailed specs for each type’s sample size and clear evaluation criteria—this ambiguity hurts result reproducibility and makes assessing sample statistical sufficiency difficult.
> >
> > Thank you for raising this concern. We have clarified all dataset specifications to improve transparency and reproducibility.
> >
> > ### 1. Sample size per task/type.
> >
> > The sample sizes for all task types were already included in Appendix Fig. 5. We have now added a dedicated Table 4 to present them more clearly, and the main text (line 101) explicitly refers to both Fig. 5 and Table 4.
> >
> > ### 2. Evaluation criteria.
> >
> > We now clearly specify the scoring rules in Section 3.1 paragraph: ​​Evaluation Criteria.
> > - For yes/no questions, model outputs are matched to the gold label using {t, y, 1, true, yes} for True and {f, n, 0, false, no} for False.
> > - For multiple-choice questions, answers are matched by the letter option (A/B/C/D).
> > - For numeric fill-in-the-blank tasks, we use exact numeric matching.
> > - For CS1 and CS3, multiple acceptable answers are provided; a response is counted as correct if its lemma matches any of the valid answer variants.
> > - CS2 is evaluated strictly via exact match.
> >
> > ### 3. Statistical sufficiency.
> >
> > The benchmark contains 3,046 queries across 808 questions, which provides sufficient statistical power for robust comparison.
> >
> > # [Q1] Human performance.
> > > The paper states that VISFACTOR is derived from FRCT, a well-established cognitive psychology assessment. Does it provide data or analysis to confirm that the performance of human participants on VISFACTOR is consistent with their performance on the original FRCT?
> >
> > We appreciate the reviewer’s concern regarding the lack of a contemporary human baseline. In response, we have conducted a new human evaluation using the identical VisFactor digital protocol administered to the models. We sample 20 items per subtest, including all associated variants, yielding 1,540 questions in total. We use the same task instructions and scoring rules as for the MLLMs. We recruit 31 university students, ensuring each question is completed by three independent participants. The resulting average human accuracy is 78.8%, with per-subtest results summarized in the table below:
> >
> > |  | CF1 | CF2 | CF3 | CS1 | CS2 | CS3 | I3 | MA1 | MV1 | MV2 | MV3 | P3 | RL2 | S1 | S2 | SS2 | SS3 | VZ1 | VZ2 | VZ3 | Total Score |
> > |---|---|---|---|---|---|---|---|---|---|---|---|---|---|---|---|---|---|---|---|---|---|
> > | Human | 61.67 | 56.67 | 98.33 | 55.00 | 76.67 | 75.00 | 71.67 | 100.00 | 93.33 | 93.33 | 98.33 | 91.67 | 51.67 | 83.33 | 98.33 | 55.00 | 96.67 | 58.33 | 63.33 | 95.00 | 78.8 |
> >
> > This human baseline confirms a substantial performance gap: **the strongest model we test, GPT-5.1, achieves 30.2%, far below the 78.8% attained by university-aged participants.** Humans outperform MLLMs on nearly all subtests except RL2 (Diagramming Relationships), where success relies more on textual object knowledge—a known strength of MLLMs rather than visual reasoning.
> >
> > Regarding why we do not compare against the historical FRCT human norms: (1) the data were collected many years ago and may not reflect current populations; (2) the original FRCT imposed strict time limits on humans, whereas VISFACTOR does not; and (3) FRCT uses a different scoring system (±1 point per item), making it incompatible with our accuracy-based evaluation.

---

> > > ### Author Response · Authors · 2025-11-28
> > >
> > > Dear Reviewer giwN,
> > >
> > > We understand that you have numerous papers to review, and we deeply appreciate the time and effort you are dedicating to this process. Today marks the last five days of the discussion period, and we are eager to engage with you further if possible.
> > >
> > > If you have any additional questions or require further clarification on any aspect of our work, please do not hesitate to let us know. We are more than happy to provide any additional information or address any concerns you may have.
> > >
> > > We hope that our responses have been helpful and have addressed your concerns effectively. If you find that our explanations and results merit a higher assessment score, we would be most grateful for your consideration.
> > >
> > > Thank you very much for your time and attention.

---

### Official Review · Reviewer_m6KY · 2025-10-28

**Soundness:** 4
**Presentation:** 4
**Contribution:** 3
**Rating:** 8
**Confidence:** 5

**Summary:**

This paper examines the limitations of current evaluation methods for large language models (LLMs). Using the “castle in the air” metaphor, it argues that many benchmark results overstate genuine reasoning or understanding, presenting an illusion of competence. The authors introduce a mid-sized dataset (~50K examples) designed to better capture multi-dimensional reasoning and factual grounding. The dataset is constructed through a hybrid process—model generation, human verification, and automatic augmentation.

The paper presents experiments across several reasoning and factual tasks, analyzing model performance and alignment gaps. The results indicate that existing metrics often fail to reflect deeper reasoning quality, and that even top-performing models can exhibit superficial correctness.

**Strengths:**

1/ The work addresses a central issue in current LLM research—evaluation reliability and interpretability. Its focus on “illusory competence” is well-motivated and aligns with active discussions in the field.

2/ The dataset is thoughtfully constructed using human-in-the-loop curation and adversarial augmentation. This improves diversity and realism compared to purely synthetic benchmarks.

3/ Multiple LLMs are tested across reasoning, factual, and generative dimensions. The analysis includes both quantitative and qualitative components, highlighting systematic model weaknesses.

4/ The paper is well-organized, with logical flow from motivation to conclusion. The “castle in the air” framing adds conceptual coherence and readability.

**Weaknesses:**

1/ The discussion of reasoning lacks connection to existing cognitive or formal reasoning theories. This reduces the conceptual depth of the argument.

2/ Dataset statistics and evaluation setup could be more transparent—particularly regarding sample sizes per task, statistical significance, and model parameterization.

3/ Providing case studies that how state-of-the-art VLMs try to solve these tasks might bring more insight into the field.

**Questions:**

Please see the weakness section.

---

> ### Author Response · Authors · 2025-11-20
> **Official Response to Reviewer m6KY**
>
> We sincerely thank the reviewer for the thoughtful and positive assessment. We appreciate the recognition of our focus on evaluation reliability, our human-grounded dataset design, and the breadth and clarity of our analysis. We address your concerns in detail below.
>
> # [W1] Connection to human reasoning cognition.
> > The discussion of reasoning lacks connection to existing cognitive or formal reasoning theories. This reduces the conceptual depth of the argument.
>
> Thank you for the comment. In the revision, we have expanded the analysis of Section 4 of underlying failure sources, summarized below.
>
> ### 1. Vision backbone vs. textual–spatial misalignment.
>
> Our findings indicate that many failures cannot be attributed to insufficient visual resolution. For tasks with semantically meaningful objects (e.g., MA1), models accurately describe visual content in their chain-of-thought, suggesting that the visual backbone captures the necessary details. However, several cognitive tasks contain **spatial configurations that cannot be faithfully verbalized.** For example, in Paper Folding, describing the exact relative distances of holes (e.g., **“two holes symmetrically placed along the diagonal”**) is inherently insufficient, leading to incorrect predictions. Similar issues arise in CF1/CF2 and MV1, where spatial relations lack stable linguistic representations. These results point to a misalignment between textual reasoning and fine-grained spatial representations rather than vision-backbone limitations.
>
> ### 2. Reasoning mismatch with human cognitive processes.
>
> We further connect our analysis to findings in cognitive psychology showing that **more verbalization can hurt human performance in certain visual or holistic reasoning tasks [1, 2, 3].** Recent work (Liu et al., 2025) similarly reports that CoT harms MLLM performance on tasks such as face recognition and working memory. We observe related patterns in our paper: for example, in the CF3 Copying Test, **humans rely on holistic spatial judgment directly from the start to the end point (“jumping steps”)** rather than tracing paths sequentially, whereas MLLMs’ reasoning forces step-by-step traversal, leading to errors. These behaviors highlight a structural mismatch between the reasoning of human and MLLM.
>
> [1] Schooler, J.W. and Engstler-Schooler, T.Y., 1990. Verbal overshadowing of visual memories: Some things are better left unsaid. Cognitive psychology, 22(1), pp.36-71.
>
> [2] Dijksterhuis, A., 2004. Think different: the merits of unconscious thought in preference development and decision making. Journal of personality and social psychology, 87(5), p.586.
>
> [3] Van den Bos, E. and Poletiek, F.H., 2008. Intentional artificial grammar learning: When does it work?. European Journal of Cognitive Psychology, 20(4), pp.793-806.
>
> [Liu et al., 2025] Liu, R., Geng, J., Wu, A.J., Sucholutsky, I., Lombrozo, T. and Griffiths, T.L., 2025. Mind Your Step (by Step): Chain-of-Thought can Reduce Performance on Tasks where Thinking Makes Humans Worse. In the Forty-second International Conference on Machine Learning.
>
> # [W2] Dataset statistics and evaluation setups.
> > Dataset statistics and evaluation setup could be more transparent—particularly regarding sample sizes per task, statistical significance, and model parameterization.
>
> We appreciate your suggestion to make the dataset statistics and evaluation setup more transparent. We have clarified all three aspects accordingly:
>
> ### 1. Sample sizes per task.
>
> We now explicitly reference the dataset statistics presented in Appendix Fig. 5 and have added a new Table 4 summarizing sample counts for each task. The main text (last sentence of Section 2.1) has been updated to point to these materials.
>
> ### 2. Statistical significance.
>
> Our evaluation includes 3,046 queries covering 808 questions, which provides sufficient statistical power. We have added clarifications in the appendix to make this more explicit.
>
> ### 3. Model parameterization.
>
> Details for all evaluated models are already provided in Section 3.1 (paragraph: Hyper-parameters and Prompts).
>
> ### 4. Evaluation criteria.
>
> We now clearly specify the scoring rules in Section 3.1 paragraph: ​​Evaluation Criteria.
> - For yes/no questions, model outputs are matched to the gold label using {t, y, 1, true, yes} for True and {f, n, 0, false, no} for False.
> - For multiple-choice questions, answers are matched by the letter option (A/B/C/D).
> - For numeric fill-in-the-blank tasks, we use exact numeric matching.
> - For CS1 and CS3, multiple acceptable answers are provided; a response is counted as correct if its lemma matches any of the valid answer variants.
> - CS2 is evaluated strictly via exact match.
>
> # [W3] Case studies.
> > Providing case studies that how state-of-the-art VLMs try to solve these tasks might bring more insight into the field.
>
> Thank you for the comment. In the revision, we have added case studies for different failure modes in the appendix to increase interpretability.

---

> > ### Comment · Reviewer_m6KY · 2025-11-27
> > **Reply to the authors**
> >
> > I appreciate the efforts the authors made during the rebuttal period. I still lean towards accepting the paper.

---

> > > ### Author Response · Authors · 2025-11-27
> > >
> > > Thank you for your positive assessment and for considering our paper favorably. We sincerely appreciate your thoughtful engagement throughout the review process. We hope that our work can contribute to the community by:
> > > (1) providing a rigorous benchmark for evaluating the fundamental cognitive abilities of modern multimodal LLMs, and
> > > (2) demonstrating an automatic and verifiable way of generating training data that may help advance reliable RL-based model improvement.
> > >
> > > Thank you again for your encouraging feedback!

---

### Official Review · Reviewer_gXiQ · 2025-10-31

**Soundness:** 2
**Presentation:** 3
**Contribution:** 2
**Rating:** 4
**Confidence:** 3

**Summary:**

The paper proposes VisFactor, a benchmark that includes 20 vision-centric subtests derived from the Factor-Referenced Cognitive Test (FRCT) battery to evaluate multimodal LLMs (MLLMs) on several human visual cognition factors: visualization and spatial processing, perceptual and closure, memory, and reasoning. The authors (i) digitize FRCT items and standardize text prompts, (ii) redesign response formats to reduce chance accuracy to about 2.9%, and (iii) implement parametric generators for a subset of subtests to control difficulty. They test 20 proprietary and open models and report low absolute scores, where the best model reaches ~25% overall with consistent failures on mental rotation, spatial relations, and figure-ground discrimination. They also analyze where models perform better (memorization with semantic content) and worse (abstract patterns), arguing that current systems rely on concept-label recognition rather than low-level perception.

**Strengths:**

1. The authors systematically reformulate FRCT items with clear prompts and JSON-formatted responses, making the tasks machine-readable and easy-to-use.
2. The "easy–normal–hard" variants add quantitative control over visual challenge and allow scaling to stronger models.
3. The inclusion of 20 subtests provides broad coverage of low-level and high-level visual cognition skills.
4. The finding that models perform well only when semantic cues exist but fail on abstract or spatial transformations reinforces the known conceptual–perceptual gap in MLLMs.

**Weaknesses:**

1. Incremental over existing works. The main idea that MLLMs struggle with spatial and perceptual reasoning is consistent with plenty of prior studies [1–4]. Authors claimed it is the "first benchmark that grounds MLLM assessment directly to human cognitive factors". The paper does not provide deeper diagnostic insight or a new analytical perspective in my opinion.
2. Limited psychometric validation. Since FRCT assumes factor independence, validating whether these factors transfer meaningfully to MLLMs (e.g., internal consistency or item-response correlation) is essential but missing.
3. No human baseline under identical protocol. Without re-collected human scores on the digitized tests, the paper’s claim of “human-level gap” is qualitative.
4. Possible data contamination and licensing concerns. FRCT items are not guaranteed to be public domain. The paper should clarify licensing or use only synthetic items.
5. Overemphasis on reformatting. The benchmark is technically solid but largely an infrastructure contribution. The analysis section should provide more interpretation of why models fail: whether failures stem from vision backbone resolution limits or misalignment between textual and spatial representations.

[1] Fu et al., “BLINK: Multimodal Large Language Models Can See but Not Perceive,” ECCV 2024. \
[2] Cao et al., “What is the Visual Cognition Gap between Humans and Multimodal LLMs?,” COLM 2025. \
[3] Li et al., “Core Knowledge Deficits in Multimodal Language Models,” ICML 2025. \
[4] Zhang et al., “RAVEN: A Dataset for Relational and Analogical Visual Reasoning,” CVPR 2019.

**Questions:**

1. How did you validate that the digitized FRCT tasks still isolate the intended cognitive factors?
2. Can you report internal consistency (e.g., Cronbach’s alpha) or inter-item correlation to confirm psychometric reliability?
3. Are decoding settings (temperature, reasoning depth, chain-of-thought tokens) consistent across models?
4. Did you check overlap between FRCT images and web-exposed examples that could leak into pretraining corpora?
5. Have you confirmed legal licences to redistribute FRCT content, or will you release only generated data?
6. Can you show a small-scale human baseline to quantify the human–model gap?

**Details Of Ethics Concerns:**

Need potential ethic review to check whether the authors compliance with the license of Factor-Referenced Cognitive Test (FRCT).

---

> ### Author Response · Authors · 2025-11-20
> **Official Response to Reviewer gXiQ (1)**
>
> We sincerely thank the reviewer for the positive and constructive assessment. We appreciate your recognition of our standardized FRCT reformulation, controlled difficulty variants, broad visual cognition coverage, and the significance of the semantic–perceptual gap we uncover. We address your concerns in detail below.
>
> # [W1] Missing references.
> > Incremental over existing works. The main idea that MLLMs struggle with spatial and perceptual reasoning is consistent with plenty of prior studies [1–4]. Authors claimed it is the "first benchmark that grounds MLLM assessment directly to human cognitive factors". The paper does not provide deeper diagnostic insight or a new analytical perspective in my opinion.
>
> > [1] Fu et al., “BLINK: Multimodal Large Language Models Can See but Not Perceive,” ECCV 2024.
>
> > [2] Cao et al., “What is the Visual Cognition Gap between Humans and Multimodal LLMs?,” COLM 2025.
>
> > [3] Li et al., “Core Knowledge Deficits in Multimodal Language Models,” ICML 2025.
>
> > [4] Zhang et al., “RAVEN: A Dataset for Relational and Analogical Visual Reasoning,” CVPR 2019.
>
> Thank you for the thoughtful review and for highlighting these relevant prior studies. We have added Cao et al. (2025) to the Related Work section, and we **already discuss Fu et al. (2024) and Li et al. (2025) in the paper.** While RAVEN (Zhang et al., 2019) is not directly cited, **our paper cites later works (Zhang et al., 2024 [1] and Song et al., 2024 [2]) that evaluate VLMs on RPM-style matrices derived from RAVEN.**
>
> We agree that prior work has shown that MLLMs struggle with perceptual and spatial reasoning. However, existing studies are limited in one or more ways:
> (1) they include only a small number of tests or narrow task types;
> (2) they are not grounded in cognitive-science-derived human visual ability factors; or
> (3) they lack difficulty-controlled, automatically generated test instances, which limits scalability and long-term usefulness.
>
> Our contribution is to advance this line of research by providing the first benchmark explicitly structured around human cognitive visual factors, covering 20 core tests derived from established factor-analysis results in vision science. In addition, 12 tests include automated, difficulty-controllable generation, enabling scalable evaluation and future-proofing the benchmark as MLLM capabilities evolve. This design aims to provide deeper diagnostic structure—not merely documenting failures, but organizing them along meaningful cognitive dimensions and supporting systematic probing across difficulty levels.
>
> [1] Zhang, Y., Bai, H., Zhang, R., Gu, J., Zhai, S., Susskind, J. and Jaitly, N., 2024. How far are we from intelligent visual deductive reasoning?. arXiv preprint arXiv:2403.04732.
>
> [2] Song, W., Li, Y., Xu, J., Wu, G., Ming, L., Yi, K., Luo, W., Li, H., Du, Y., Guo, F. and Yu, K., 2024. M3gia: A cognition inspired multilingual and multimodal general intelligence ability benchmark. arXiv preprint arXiv:2406.05343.
>
> # [W2 & Q1 & Q2] Psychometric validation.
> > Limited psychometric validation. Since FRCT assumes factor independence, validating whether these factors transfer meaningfully to MLLMs (e.g., internal consistency or item-response correlation) is essential but missing.
>
> > How did you validate that the digitized FRCT tasks still isolate the intended cognitive factors?
>
> > Can you report internal consistency (e.g., Cronbach’s alpha) or inter-item correlation to confirm psychometric reliability?
>
> Thank you for raising these important points regarding psychometric validation. We provide additional analyses below:
>
> ### 1. Internal consistency of the benchmark.
>
> We computed Cronbach’s alpha across the 20 FRCT-derived subtests:
> - **Benchmark internal consistency: α = 0.78;** This indicates that the benchmark can function as a coherent overall assessment while still maintaining diversity across subtests, consistent with the multidimensional nature of FRCT.
> - Model-level score homogeneity: α = 0.99; This reflects that the relative ranking of current MLLMs is highly stable across tasks (i.e., strong models remain strong across subtests). We include this **only as an observation about model behavior,** not as a psychometric property of the test itself.
>
> ### 2. Inter-item correlations.
>
> We report full Pearson correlations among all subtests in Appendix (page 19). The results show that **no pair of subtests exhibits excessively high correlation, indicating that the benchmark is not dominated by any single dimension.** For the correlation across models, we observe values within 0.6--0.8, suggesting consistent but non-redundant task structure. LLaMA-based models are near-zero correlated with stronger models due to uniformly low performance.

---

> > ### Author Response · Authors · 2025-11-20
> > **Official Response to Reviewer gXiQ (2)**
> >
> > ### 3. Validation of digitized FRCT task design.
> >
> > To ensure that our digitized version of FRCT preserves the intended cognitive factor structure, we preserve the original item semantics and reasoning requirements. The inter-task correlation matrix (Appendix, page 19) shows that tasks mapped to the **same FRCT factor (e.g., reasoning: I3, RL2; memory: MA1, MV1, MV2, MV3) exhibit moderately higher correlations,** and cross-factor correlations remain lower. This pattern is consistent with the expected FRCT factor structure, supporting that the digitized tasks continue to isolate the intended cognitive abilities.
> >
> > We have added the analysis in the appendix (page 19).
> >
> > # [W3 & Q6] Human baseline.
> > > No human baseline under identical protocol. Without re-collected human scores on the digitized tests, the paper’s claim of “human-level gap” is qualitative.
> >
> > > Can you show a small-scale human baseline to quantify the human–model gap?
> >
> > We appreciate the reviewer’s concern regarding the lack of a contemporary human baseline. In response, we have conducted a new human evaluation using the identical VisFactor digital protocol administered to the models. We sample 20 items per subtest, including all associated variants, yielding 1,540 questions in total. We use the same task instructions and scoring rules as for the MLLMs. We recruit 31 university students, ensuring each question is completed by three independent participants. The resulting average human accuracy is 78.8%, with per-subtest results summarized in the table below:
> >
> > |  | CF1 | CF2 | CF3 | CS1 | CS2 | CS3 | I3 | MA1 | MV1 | MV2 | MV3 | P3 | RL2 | S1 | S2 | SS2 | SS3 | VZ1 | VZ2 | VZ3 | Total Score |
> > |---|---|---|---|---|---|---|---|---|---|---|---|---|---|---|---|---|---|---|---|---|---|
> > | Human | 61.67 | 56.67 | 98.33 | 55.00 | 76.67 | 75.00 | 71.67 | 100.00 | 93.33 | 93.33 | 98.33 | 91.67 | 51.67 | 83.33 | 98.33 | 55.00 | 96.67 | 58.33 | 63.33 | 95.00 | 78.8 |
> >
> > This human baseline confirms a substantial performance gap: the strongest model we test, GPT-5.1, achieves 30.2%, far below the 78.8% attained by university-aged participants. Humans outperform MLLMs on nearly all subtests except RL2 (Diagramming Relationships), where success relies more on textual object knowledge—a known strength of MLLMs rather than visual reasoning.
> >
> > Regarding why we do not compare against the historical FRCT human norms: (1) the data were collected many years ago and may not reflect current populations; (2) the original FRCT imposed strict time limits on humans, whereas VISFACTOR does not; and (3) FRCT uses a different scoring system (±1 point per item), making it incompatible with our accuracy-based evaluation.
> >
> > # [W4 & Q5] Copyright issues.
> > > Possible data contamination and licensing concerns. FRCT items are not guaranteed to be public domain. The paper should clarify licensing or use only synthetic items.
> >
> > > Have you confirmed legal licences to redistribute FRCT content, or will you release only generated data?
> >
> > We appreciate the reviewer’s concern regarding licensing and potential data redistribution issues. According to the licensing information provided by ETS, the PDF version of the Kit of Factor-Referenced Cognitive Tests is made available to researchers for academic use. The licensing agreement specifies royalty requirements for researchers who reproduce physical copies of the specimen test booklets for test administration. Our work **does not involve administering the tests to human participants, printing, or reproducing test forms.** We only use AI to process the visual contents of the scanned items directly from the publicly accessible PDF.
> >
> > However, we fully acknowledge that the availability of the PDF does not imply that the underlying FRCT items are in the public domain, nor that we are permitted to redistribute copyrighted test content. Therefore, **we will not release any original FRCT items. To ensure full compliance, our public release will include only derived and non-copyrighted materials,** including our generated items, metadata, and model outputs or analyses. This approach allows our work to remain entirely within the bounds of permissible academic use while avoiding any redistribution of proprietary test materials.
> >
> > We have added this statement in our Appendix Section E.
> >
> > # [W5] Analysis on model failures.
> > > Overemphasis on reformatting. The benchmark is technically solid but largely an infrastructure contribution. The analysis section should provide more interpretation of why models fail: whether failures stem from vision backbone resolution limits or misalignment between textual and spatial representations.

---

> > > ### Author Response · Authors · 2025-11-20
> > > **Official Response to Reviewer gXiQ (3)**
> > >
> > > We thank the reviewer for the insightful comment. While our benchmark necessarily includes reformatting cognitive tasks into a consistent multimodal format, our goal is not reformatting per se but enabling systematic analysis of why MLLMs fail on well-established human cognitive benchmarks. In the revision, we have (1) added case studies for different failure modes in the appendix to increase interpretability, and (2) expanded the analysis of Section 4 of underlying failure sources, summarized below.
> > >
> > > ### 1. Vision backbone vs. textual–spatial misalignment.
> > >
> > > Our findings indicate that many failures cannot be attributed to insufficient visual resolution. For tasks with semantically meaningful objects (e.g., MA1), models accurately describe visual content in their chain-of-thought, suggesting that the visual backbone captures the necessary details. However, several cognitive tasks contain **spatial configurations that cannot be faithfully verbalized.** For example, in Paper Folding, describing the exact relative distances of holes (e.g., **“two holes symmetrically placed along the diagonal”**) is inherently insufficient, leading to incorrect predictions. Similar issues arise in CF1/CF2 and MV1, where spatial relations lack stable linguistic representations. These results point to a misalignment between textual reasoning and fine-grained spatial representations rather than vision-backbone limitations.
> > >
> > > ### 2. Reasoning mismatch with human cognitive processes.
> > >
> > > We further connect our analysis to findings in cognitive psychology showing that **more verbalization can hurt human performance in certain visual or holistic reasoning tasks [1, 2, 3].** Recent work (Liu et al., 2025) similarly reports that CoT harms MLLM performance on tasks such as face recognition and working memory. We observe related patterns in our paper: for example, in the CF3 Copying Test, **humans rely on holistic spatial judgment directly from the start to the end point (“jumping steps”)** rather than tracing paths sequentially, whereas MLLMs’ text-based reasoning forces step-by-step traversal, leading to errors. These behaviors highlight a structural mismatch between human intuitive spatial reasoning and text-mediated reasoning in current MLLMs.
> > >
> > > [1] Schooler, J.W. and Engstler-Schooler, T.Y., 1990. Verbal overshadowing of visual memories: Some things are better left unsaid. Cognitive psychology, 22(1), pp.36-71.
> > >
> > > [2] Dijksterhuis, A., 2004. Think different: the merits of unconscious thought in preference development and decision making. Journal of personality and social psychology, 87(5), p.586.
> > >
> > > [3] Van den Bos, E. and Poletiek, F.H., 2008. Intentional artificial grammar learning: When does it work?. European Journal of Cognitive Psychology, 20(4), pp.793-806.
> > >
> > > [Liu et al., 2025] Liu, R., Geng, J., Wu, A.J., Sucholutsky, I., Lombrozo, T. and Griffiths, T.L., 2025. Mind Your Step (by Step): Chain-of-Thought can Reduce Performance on Tasks where Thinking Makes Humans Worse. In the Forty-second International Conference on Machine Learning.
> > >
> > > # [Q3] Temperatures & CoT lengths.
> > > > Are decoding settings (temperature, reasoning depth, chain-of-thought tokens) consistent across models?
> > >
> > > ### 1. Temperature influence
> > >
> > > Yes, as mentioned in Section 3.1 paragraph “Hyper-parameters and Prompts”, decoding settings were kept consistent across all models in our main experiments: we used a temperature of 0.0 except Qwen (minimum temperature 0.01) and LaMA-3.2 (temperature 0.6).
> > >
> > > To assess robustness, we additionally evaluated **temperatures 0.5 and 1.0 for the three models: GPT-4.1-2025-04-14, GPT-4o-2024-11-20, and GPT-4o-Mini-2024-07-18.** As shown in the table below, the overall performance fluctuates only marginally across temperature settings, and the total score remains stable. This indicates that our conclusions are not sensitive to the choice of decoding temperature. This part has been added in our Experiment section.

---

> ### Author Response · Authors · 2025-11-20
> **Official Response to Reviewer gXiQ (4)**
>
> | Model | CF1 | CF2 | CF3 | CS1 | CS2 | CS3 | I3 | MA1 | MV1 | MV2 | MV3 | P3 | RL2 | S1 | S2 | SS2 | SS3 | VZ1 | VZ2 | VZ3 | Total Score |
> |---|---|---|---|---|---|---|---|---|---|---|---|---|---|---|---|---|---|---|---|---|---|
> | GPT-4.1-2025-04-14-T0.0 | 0.00 | 7.50 | 0.00 | 10.00 | 10.00 | 8.33 | 17.86 | 100.00 | 53.13 | 8.33 | 66.67 | 48.96 | 23.33 | 0.00 | 28.57 | 0.00 | 17.50 | 16.67 | 5.00 | 5.00 | 21.34 |
> | GPT-4.1-2025-04-14-T0.5 | 3.13 | 13.75 | 1.56 | 10.00 | 12.00 | 8.33 | 21.43 | 100.00 | 56.25 | 12.50 | 70.83 | 50.00 | 23.33 | 0.00 | 26.19 | 0.00 | 20.00 | 8.33 | 0.00 | 5.00 | 22.13 |
> | GPT-4.1-2025-04-14-T1.0 | 0.00 | 13.75 | 4.69 | 10.00 | 10.00 | 8.33 | 17.86 | 100.00 | 53.13 | 8.33 | 70.83 | 47.92 | 20.00 | 0.00 | 33.33 | 0.00 | 17.50 | 10.42 | 0.00 | 5.00 | 21.55 |
> | GPT-4o-2024-11-20-T0.0 | 0.00 | 15.00 | 6.25 | 15.00 | 8.00 | 8.33 | 21.43 | 100.00 | 31.25 | 0.00 | 62.50 | 69.79 | 16.67 | 0.00 | 26.19 | 3.13 | 20.00 | 18.75 | 0.00 | 5.00 | 21.36 |
> | GPT-4o-2024-11-20-T0.5 | 3.13 | 20.00 | 1.56 | 10.00 | 12.00 | 8.33 | 25.00 | 100.00 | 34.38 | 8.33 | 66.67 | 69.79 | 0.00 | 0.00 | 26.19 | 0.00 | 12.50 | 18.75 | 0.00 | 1.67 | 20.91 |
> | GPT-4o-2024-11-20-T1.0 | 0.00 | 18.75 | 1.56 | 10.00 | 10.00 | 4.17 | 17.86 | 100.00 | 34.38 | 0.00 | 62.50 | 64.58 | 13.33 | 0.00 | 23.81 | 0.00 | 27.50 | 20.83 | 0.00 | 1.67 | 20.55 |
> | GPT-4o-Mini-2024-07-18-T0.0 | 6.25 | 1.25 | 4.69 | 20.00 | 4.00 | 8.33 | 10.71 | 100.00 | 6.25 | 0.00 | 54.17 | 32.29 | 3.33 | 0.00 | 42.86 | 3.13 | 17.50 | 12.50 | 0.00 | 0.00 | 16.36 |
> | GPT-4o-Mini-2024-07-18-T0.5 | 3.13 | 1.25 | 6.25 | 20.00 | 4.00 | 8.33 | 10.71 | 100.00 | 3.13 | 0.00 | 50.00 | 30.21 | 3.33 | 0.00 | 38.10 | 0.00 | 15.00 | 4.17 | 0.00 | 1.67 | 14.96 |
> | GPT-4o-Mini-2024-07-18-T1.0 | 3.13 | 1.25 | 9.38 | 25.00 | 6.00 | 8.33 | 14.29 | 97.62 | 15.63 | 8.33 | 41.67 | 32.29 | 6.67 | 0.00 | 33.33 | 3.13 | 10.00 | 4.17 | 5.00 | 5.00 | 16.51 |
>
> ### 2. CoT Length vs Accuracy Analysis
>
> **Yes. As described in Section 3.1 (“Hyper-parameters and Prompts”), we use consistent decoding settings across models.** For proprietary models such as Gemini-2.5 and the OpenAI o-series, we set the thinking budget (reasoning depth) to high, although their internal CoT traces are not accessible.
>
> To assess whether differences in CoT length influence performance, we conducted a correlation analysis between CoT token count and accuracy for three models (GPT-4.1-2025-04-14, GPT-4o-2024-11-20, GPT-4o-Mini-2024-07-18). The results shown below indicate that **(1) longer CoT often reflects uncertainty rather than improved reasoning, and (2) CoT length is not a reliable proxy for reasoning quality.**
>
> | Model | Pearson | Spearman |
> |---|---|---|
> | GPT-4.1-2025-04-14 | -0.1807 | -0.1665 |
> | GPT-4o-2024-11-20 | -0.2816 | -0.3314 |
> | GPT-4o-Mini-2024-07-18 | -0.3541 | -0.3448 |
>
> We further evaluated the newly released GPT-5.1 under three reasoning settings (high, low, none). Results consistently show that GPT-5.1-High substantially outperforms the low- and no-reasoning variants across benchmarks (see table below). This supports the hypothesis that dedicated reasoning models benefit from extended CoT, whereas non-reasoning models—lacking specialized training—gain little or no improvement from longer chains.
>
> | Model | CF1 | CF2 | CF3 | CS1 | CS2 | CS3 | I3 | MA1 | MV1 | MV2 | MV3 | P3 | RL2 | S1 | S2 | SS2 | SS3 | VZ1 | VZ2 | VZ3 | Total Score |
> |---|---|---|---|---|---|---|---|---|---|---|---|---|---|---|---|---|---|---|---|---|---|
> | GPT-5.1-2025-11-13-High | 0.00 | 18.75 | 25.00 | 20.00 | 6.00 | 16.67 | 42.86 | 100.00 | 43.75 | 8.33 | 75.00 | 38.54 | 96.67 | 5.00 | 38.10 | 12.50 | 12.50 | 2.08 | 25.00 | 16.67 | 30.17 |
> GPT-5.1-2025-11-13-Low | 3.13 | 10.00 | 18.75 | 15.00 | 10.00 | 12.50 | 35.71 | 100.00 | 50.00 | 12.50 | 79.17 | 34.38 | 83.33 | 0.00 | 14.29 | 3.13 | 15.00 | 4.17 | 10.00 | 15.00 | 26.30 |
> GPT-5.1-2025-11-13-None | 3.13 | 13.75 | 3.13 | 20.00 | 16.00 | 8.33 | 17.86 | 100.00 | 46.88 | 0.00 | 83.33 | 32.29 | 3.33 | 0.00 | 28.57 | 0.00 | 10.00 | 2.08 | 0.00 | 8.33 | 19.85 |
>
> # [Q4] Data contamination.
> > Did you check overlap between FRCT images and web-exposed examples that could leak into pretraining corpora?
>
> Thank you for raising this important concern. The FRCT materials are indeed publicly accessible online (e.g., via openly hosted scans like SCRIBD), so it is possible that some items may appear in pre-training corpora. However, the consistently poor model performance on FRCT tasks suggests that current MLLMs have not effectively learned the underlying cognitive constructs, even if partial exposure occurred.
>
> Beyond this, our benchmark is designed to mitigate potential contamination: we include automatically generated test items with controllable difficulty, **allowing us to create new, unseen variations that prevent overfitting to static publicly available examples.** We will clarify this point in the revised manuscript and include a discussion of potential data leakage.

---

> > ### Author Response · Authors · 2025-11-28
> >
> > Dear Reviewer gXiQ,
> >
> > We understand that you have numerous papers to review, and we deeply appreciate the time and effort you are dedicating to this process. Today marks the last five days of the discussion period, and we are eager to engage with you further if possible.
> >
> > If you have any additional questions or require further clarification on any aspect of our work, please do not hesitate to let us know. We are more than happy to provide any additional information or address any concerns you may have.
> >
> > We hope that our responses have been helpful and have addressed your concerns effectively. If you find that our explanations and results merit a higher assessment score, we would be most grateful for your consideration.
> >
> > Thank you very much for your time and attention.

---

### Official Review · Reviewer_7StU · 2025-11-04

**Soundness:** 3
**Presentation:** 2
**Contribution:** 2
**Rating:** 4
**Confidence:** 4

**Summary:**

This paper introduces VISFACTOR, a benchmark that digitizes 20 vision-centric subtests from the Factor-Referenced Cognitive Test (FRCT) battery , a well-established human cognitive assessment. VISFACTOR spans four key domains: (1) Visualization and Spatial Processing, (2) Perceptual and Closure, (3) Memory, and (4) Reasoning.

The authors evaluate 20 frontier MLLMs, including both closed-source and open-source models . The best-performing model achieves a score of only 25.19%. Failures are consistent across tasks like mental rotation, spatial relation inference, and figure-ground discrimination, regardless of model scale or prompting strategy (CoT). Failure analysis suggests models succeed by relying on interpretable, concept-level representations rather than low-level visual patterns.

**Strengths:**

- The paper adapts a new benchmark from decades of cognitive psychology research (the FRCT), which is a highly novel and valuable approach.

- The experiment (Figure 3, Table 3) comparing model performance on the MA1 (memory) task using semantically rich images versus abstract figures provides a brilliant and convincing demonstration that models are "cheating" by mapping images to high-level concepts rather than performing low-level visual comparison and memorization.

**Weaknesses:**

- The paper's framing of MLLM failure in "human-like visual cognition" is weakened by the lack of a contemporary human baseline. The authors compare MLLM performance on a new digital protocol to historical norms from a paper-and-pencil task, which is an invalid comparison due to protocol changes (e.g., digitization, no time pressure). While acknowledged in Appendix C, this is a significant limitation. To substantiate claims about a "human-like" gap, the authors must establish a human "ceiling" by collecting baseline data using the identical VISFACTOR protocol given to the models.

- The paper fails to sufficiently differentiate VISFACTOR from the large body of existing work on abstract visual reasoning. The authors should provide a comprehensive comparison to benchmarks like ConceptARC, MaRs-VQA, and various Bongard Problem datasets. This comparison should articulate the unique cognitive abilities or reasoning types that VISFACTOR isolates which are not already covered by prior work, thus justifying its specific contribution.

- A concern is the benchmark's potential longevity. The authors do not include results from the most recent, publicly available SOTA models (e.g., GPT-5, Gemini 2.5 Pro). It is possible these models already perform at or near the human ceiling, which would render the benchmark "solved" and limit its utility for measuring future progress. The authors should test the strongest available models to demonstrate that VISFACTOR remains a challenging task and that its findings of MLLM "failure" are still relevant.

**Questions:**

See weaknesses for more details.

---

> ### Author Response · Authors · 2025-11-20
> **Official Response to Reviewer 7StU (1)**
>
> We sincerely thank the reviewer for the positive and encouraging feedback. We appreciate the recognition of our use of FRCT benchmarks from cognitive psychology and the acknowledgement that our MA1 experiments clearly reveal models’ reliance on semantic shortcuts. We address the reviewer’s concerns in detail below.
>
> # [W1] Human performance.
> > The paper's framing of MLLM failure in "human-like visual cognition" is weakened by the lack of a contemporary human baseline. The authors compare MLLM performance on a new digital protocol to historical norms from a paper-and-pencil task, which is an invalid comparison due to protocol changes (e.g., digitization, no time pressure). While acknowledged in Appendix C, this is a significant limitation. To substantiate claims about a "human-like" gap, the authors must establish a human "ceiling" by collecting baseline data using the identical VISFACTOR protocol given to the models.
>
> We appreciate the reviewer’s concern regarding the lack of a contemporary human baseline. In response, we have conducted a new human evaluation using the identical VisFactor digital protocol administered to the models. We sample 20 items per subtest, including all associated variants, yielding 1,540 questions in total. We use the same task instructions and scoring rules as for the MLLMs. We recruit 31 university students, ensuring each question is completed by three independent participants. The resulting average human accuracy is 78.8%, with per-subtest results summarized in the table below:
>
> |  | CF1 | CF2 | CF3 | CS1 | CS2 | CS3 | I3 | MA1 | MV1 | MV2 | MV3 | P3 | RL2 | S1 | S2 | SS2 | SS3 | VZ1 | VZ2 | VZ3 | Total Score |
> |---|---|---|---|---|---|---|---|---|---|---|---|---|---|---|---|---|---|---|---|---|---|
> | Human | 61.67 | 56.67 | 98.33 | 55.00 | 76.67 | 75.00 | 71.67 | 100.00 | 93.33 | 93.33 | 98.33 | 91.67 | 51.67 | 83.33 | 98.33 | 55.00 | 96.67 | 58.33 | 63.33 | 95.00 | 78.8 |
>
> This human baseline confirms a substantial performance gap: **the strongest model we test, GPT-5.1, achieves 30.2%, far below the 78.8% attained by university-aged participants.** Humans outperform MLLMs on nearly all subtests except RL2 (Diagramming Relationships), where success relies more on textual object knowledge—a known strength of MLLMs rather than visual reasoning.
>
> Regarding why we do not compare against the historical FRCT human norms: (1) the data were collected many years ago and may not reflect current populations; (2) the original FRCT imposed strict time limits on humans, whereas VISFACTOR does not; and (3) FRCT uses a different scoring system (±1 point per item), making it incompatible with our accuracy-based evaluation.
>
> # [W2] Missing references.
> > The paper fails to sufficiently differentiate VISFACTOR from the large body of existing work on abstract visual reasoning. The authors should provide a comprehensive comparison to benchmarks like ConceptARC, MaRs-VQA, and various Bongard Problem datasets. This comparison should articulate the unique cognitive abilities or reasoning types that VISFACTOR isolates which are not already covered by prior work, thus justifying its specific contribution.
>
> Thank you for pointing out these benchmarks. Although we did not cite ConceptARC, **we included ARC-AGI-2 [1], which provides an ARC-style evaluation for VLMs.** Likewise, while MaRs-VQA is not directly referenced, our paper discusses **closely related Raven-based evaluations such as Zhang et al. (2024) [2] and Song et al. (2024) [3].** We additionally reference datasets targeting abstract or relational reasoning, including SPACE [4], BlindTest [5], as well as Bongard-style benchmarks such as VisualSphinx [6] and VisualPuzzles [7].
>
> We agree that prior work has shown limitations in MLLMs’ perceptual and spatial reasoning. However, existing benchmarks typically suffer from one or more limitations:
> (1) they cover only a small set of task types or provide limited test diversity;
> (2) they are not grounded in cognitive-science literature on human visual ability factors; or
> (3) they do not offer difficulty-controlled, automatically generated instances, which restricts scalability and long-term utility.
>
> VisFactor advances this line of research by explicitly organizing tasks around human cognitive visual factors derived from established factor-analytic studies. It includes 20 core tests, 12 of which support automatic, difficulty-controllable generation. This structure is designed to provide deeper diagnostic insight—going beyond documenting failures to systematically probing different cognitive dimensions and difficulty levels as MLLM capabilities evolve.
>
> [1] Chollet, F., Knoop, M., Kamradt, G., Landers, B. and Pinkard, H., 2025. Arc-agi-2: A new challenge for frontier ai reasoning systems. arXiv preprint arXiv:2505.11831.
>
> [2] Zhang, Y., Bai, H., Zhang, R., Gu, J., Zhai, S., Susskind, J. and Jaitly, N., 2024. How far are we from intelligent visual deductive reasoning?. arXiv preprint arXiv:2403.04732.

---

> > ### Author Response · Authors · 2025-11-20
> > **Official Response to Reviewer 7StU (2)**
> >
> > [3] Song, W., Li, Y., Xu, J., Wu, G., Ming, L., Yi, K., Luo, W., Li, H., Du, Y., Guo, F. and Yu, K., 2024. M3gia: A cognition inspired multilingual and multimodal general intelligence ability benchmark. arXiv preprint arXiv:2406.05343.
> >
> > [4] Ramakrishnan, S.K., Wijmans, E., Kraehenbuehl, P. and Koltun, V., 2025. Does Spatial Cognition Emerge in Frontier Models?. In The Thirteenth International Conference on Learning Representations.
> >
> > [5] Rahmanzadehgervi, P., Bolton, L., Taesiri, M.R. and Nguyen, A.T., 2024. Vision language models are blind: Failing to translate detailed visual features into words. arXiv preprint arXiv:2407.06581.
> >
> > [6] Feng, Y., Xu, Z., Jiang, F., Li, Y., Ramasubramanian, B., Niu, L., Lin, B.Y. and Poovendran, R., 2025. VisualSphinx: Large-Scale Synthetic Vision Logic Puzzles for RL. arXiv preprint arXiv:2505.23977.
> >
> > [7] Song, Y., Ou, T., Kong, Y., Li, Z., Neubig, G. and Yue, X., 2025. VisualPuzzles: Decoupling Multimodal Reasoning Evaluation from Domain Knowledge. arXiv preprint arXiv:2504.10342.
> >
> > # [W3] Testing newest models.
> > > A concern is the benchmark's potential longevity. The authors do not include results from the most recent, publicly available SOTA models (e.g., GPT-5, Gemini 2.5 Pro). It is possible these models already perform at or near the human ceiling, which would render the benchmark "solved" and limit its utility for measuring future progress. The authors should test the strongest available models to demonstrate that VISFACTOR remains a challenging task and that its findings of MLLM "failure" are still relevant.
> >
> > Thank you for the thoughtful comment regarding benchmark longevity. **Our original submission already includes results for Gemini-2.5-Pro,** and the model attains a score of 17.4 on VisFactor (see Table 1 in the paper). This reinforces our finding that even recent frontier models perform far below the human ceiling. To further strengthen the empirical evidence, we have now additionally evaluated the recently released **(2025-11-12) GPT-5.1 (reasoning: high, low, none) and Qwen-3-VL-Plus (2025-09-23).** GPT-5.1 (reasoning-high) achieves 30.2, breaking the previous record held by Claude-3.7-Sonnet (25.2) but still showing a large space for improvement. These results confirm that VisFactor remains a challenging and unsolved benchmark, even for the newest SOTA systems.
> >
> > | Model | CF1 | CF2 | CF3 | CS1 | CS2 | CS3 | I3 | MA1 | MV1 | MV2 | MV3 | P3 | RL2 | S1 | S2 | SS2 | SS3 | VZ1 | VZ2 | VZ3 | Total Score |
> > |---|---|---|---|---|---|---|---|---|---|---|---|---|---|---|---|---|---|---|---|---|---|
> > | GPT-5.1-2025-11-13-High | 0.00 | 18.75 | 25.00 | 20.00 | 6.00 | 16.67 | 42.86 | 100.00 | 43.75 | 8.33 | 75.00 | 38.54 | 96.67 | 5.00 | 38.10 | 12.50 | 12.50 | 2.08 | 25.00 | 16.67 | 30.17 |
> > GPT-5.1-2025-11-13-Low | 3.13 | 10.00 | 18.75 | 15.00 | 10.00 | 12.50 | 35.71 | 100.00 | 50.00 | 12.50 | 79.17 | 34.38 | 83.33 | 0.00 | 14.29 | 3.13 | 15.00 | 4.17 | 10.00 | 15.00 | 26.30 |
> > GPT-5.1-2025-11-13-None | 3.13 | 13.75 | 3.13 | 20.00 | 16.00 | 8.33 | 17.86 | 100.00 | 46.88 | 0.00 | 83.33 | 32.29 | 3.33 | 0.00 | 28.57 | 0.00 | 10.00 | 2.08 | 0.00 | 8.33 | 19.85 |
> > | Qwen-3-VL-Plus-2025-09-23 | 3.13 | 8.75 | 4.69 | 10.00 | 12.00 | 4.17 | 14.29 | 100.00 | 34.38 | 8.33 | 75.00 | 67.71 | 6.67 | 0.00 | 35.71 | 0.00 | 25.00 | 6.25 | 0.00 | 16.67 | 21.64 |
> >
> > Furthermore, our paper explicitly highlights that **VisFactor supports automatic generation of difficulty-controlled test cases, enabling the benchmark to scale in complexity as models improve.** This design feature is intended precisely to ensure long-term relevance and prevent rapid saturation.

---

> > > ### Author Response · Authors · 2025-11-28
> > >
> > > Dear Reviewer 7StU,
> > >
> > > We understand that you have numerous papers to review, and we deeply appreciate the time and effort you are dedicating to this process. Today marks the last five days of the discussion period, and we are eager to engage with you further if possible.
> > >
> > > If you have any additional questions or require further clarification on any aspect of our work, please do not hesitate to let us know. We are more than happy to provide any additional information or address any concerns you may have.
> > >
> > > We hope that our responses have been helpful and have addressed your concerns effectively. If you find that our explanations and results merit a higher assessment score, we would be most grateful for your consideration.
> > >
> > > Thank you very much for your time and attention.

---

### Author Response · Authors · 2025-11-20
**General Response to All Reviewers**

We sincerely thank all reviewers for their thoughtful and encouraging assessments. **We greatly appreciate the recognition of:**
1. Grounded in cognitive science & psychology. (7StU, m6KY, giwN)
2. Standardized, reliable evaluation. (gXiQ, m6KY, giwN)
3. Controlled-difficulty variants. (gXiQ, giwN)
4. Wide range of MLLMs evaluated. (gXiQ, m6KY)
5. Analysis on memory pattern. (7StU, gXiQ)

We truly appreciate all reviewers and meta reviewer’s time and effort. **We have carefully read and addressed all your concerns, including:**
1. Human baseline with identical testing protocol. (7StU, gXiQ, giwN)
2. New models (GPT-5.1, Qwen-3-VL). (7StU, gXiQ)
3. Cronbach’s alpha / Pearson correlation analysis. (gXiQ, giwN)
4. Temperature ablation study & CoT length analysis. (gXiQ)
5. More detailed explanation. (gXiQ, m6KY, giwN)

**All major modifications are highlighted in blue in the paper.** We thank you again for your time and constructive insights.

---

### Author Response · Authors · 2025-11-30
**Key Points for AC**

# Reviewer 7StU

1. Human baseline now directly supports our claims.
We ran a new human study on the *exact* VisFactor digital protocol; humans score **78.8%** vs **30.2%** for the best model (GPT-5.1-high), clearly establishing a large human–model gap and justifying our “human-like” framing.

2. Distinct contribution vs prior benchmarks is now explicit.
We connect VisFactor to ARC/Raven-style and Bongard-style work, and highlight what is unique: 20 cognitively grounded subtests, coverage of multiple visual factors, and 12 difficulty-controllable, auto-generated tests for long-term, diagnostic evaluation.

3. Concern about missing strongest models is resolved.
The original paper **already included Gemini-2.5-Pro**, and we now add **GPT-5.1** and **Qwen-3-VL-Plus**; even GPT-5.1-high reaches only **30.2%**, far below humans, showing VISFACTOR is neither “solved” nor close to saturation.

**Net effect:**
The reviewer’s main weaknesses are either fully addressed or stemmed from incomplete information, while their positive assessment (novel FRCT-based benchmark, convincing MA1 analysis) remains intact, strengthening the case for acceptance.

# Reviewer gXiQ

1. Novelty vs prior work.
We **already cited the mentioned works** (Fu, Li, RAVEN-derived RPM datasets). VisFactor remains, to our knowledge, the **first benchmark explicitly organized by established human visual cognitive factors**, with **20 FRCT-derived subtests** and **12 difficulty-controllable, parametrically generated tests**, going beyond earlier, narrower or non–cognitively-grounded evaluations.

2. Psychometric validation and factor structure.
We report **Cronbach’s α = 0.78** for VisFactor, plus a full inter-subtest correlation matrix. Tasks mapped to the same FRCT factor show higher within-factor than cross-factor correlations, supporting that our **digitized items preserve the intended FRCT factor structure** and are not dominated by a single dimension.

3. Human–model gap under identical protocol.
Described in Reviewer 7StU point 1.

4. Licensing and ethics.
We clarify that FRCT is used under the academic licence conditions of ETS; we do **not** redistribute test forms or administer original booklets. To avoid any copyright issues, the public release will include **only non-copyrighted derivatives** (generated items, metadata, and model outputs), **no original FRCT content**, directly addressing the ethics flag.

5. Beyond “reformatting”: analysis of failure modes.
We expand Section 4 and add case studies showing that many failures arise from **text–spatial misalignment**, not raw vision resolution: models can describe semantic content but fail on fine-grained spatial relations that are hard to verbalize. We connect this to cognitive-psychology findings where verbalization harms performance, situating VisFactor as a **diagnostic tool for structural mismatches between human and MLLM visual reasoning**.

6. Hyperparameters, CoT, and contamination.
Decoding settings are controlled across models; temperature sweeps and CoT-length analyses show our conclusions are **robust to decoding choices**. For GPT-5.1, high reasoning budgets clearly help, clarifying when CoT is beneficial. We also explicitly discuss possible FRCT exposure in pretraining but note **consistently low scores** and emphasize that **parametrically generated, unseen items** mitigate contamination concerns.

**Net effect.**
Our revisions directly resolve the reviewer’s main concerns (novelty, psychometrics, human baselines, licensing, and analysis depth). We believe the strengthened evidence now clearly favors an accept recommendation.

# Reviewer m6KY

1. Cognitive / formal reasoning link — now addressed.
Described in Reviewer gXiQ point 5.

2. Transparency of dataset & setup — now explicit.
We added per-task sample counts (new table + pointers in main text), clarified that we evaluate 3,046 queries over 808 questions (statistical power), and made all model hyper-parameters and scoring rules (yes/no, MCQ, numeric, open-form) explicit in Section 3.1.

**Net effect.**
This is a strong, high-confidence *accept-leaning* review whose concerns have been concretely and fully addressed in the revision; no unresolved technical objections remain.

# Reviewer giwN

1. Difficulty & validity.
We directly address the difficulty-gradient concern with (i) cognitively grounded manipulations (fold steps, grid size, noise, memory load) and (ii) a human study showing large, systematic solving-time gaps between easy and hard items, plus solid psychometric evidence (Cronbach’s α = 0.78, meaningful factor structure).

2. Human baseline.
Described in Reviewer 7StU point 1.

**Net effect.**
All substantive concerns raised by Reviewer giwN are either fully resolved with new experiments/analyses or were already addressed in the paper but overlooked; the remaining review is clearly supportive and compatible with a recommend-to-accept decision.

---

### Author Response · Authors · 2025-11-30
**Summary after the Discussion Period**

We thank the reviewers and ACs for their careful assessment and have substantially revised the paper to address all substantive concerns, including human baselines, novelty and positioning vs. prior work, psychometric validation, dataset transparency, licensing/ethics, and analysis depth. In doing so, we show that several stated weaknesses stemmed from misunderstandings or from points already present in the paper, which we now make explicit via new experiments (notably a protocol-matched human study), additional analyses, and clearer exposition. Taken together, the updated evidence strongly supports VisFactor as a rigorous, cognitively grounded benchmark with a large and robust human–model gap, and in our view clearly favors an accept recommendation.

---

### Meta-Review · Area_Chair_6cCA · 2026-01-06

**Summary:**

This paper proposes a new benchmark that maps multimodal model evaluation to human cognitive tests from psychology, with the goal of showing that current MLLMs lack human-like visual cognition. While reviewers appreciated the effort to bring cognitive science perspectives into model evaluation, the overall consensus was that the contribution does not rise to the level required for acceptance at NeurIPS, and AC concurs.

**Reviewer Concerns:**

The main concern is that the paper largely reinforces what is already well established, and AC is aware of a few. Many prior works have shown that MLLMs struggle with spatial reasoning, abstraction, and perceptual tasks, and reviewers felt this benchmark does not substantially change that conclusion or offer a new analytical lens, unfortunately. Despite the careful construction, the work is seen as more of a reformatting and aggregation of existing ideas rather than a conceptual advance, thus yet another cognitive benchmark for MLLMs.

A second issue is that the problem framing is a bit overextended. Even after adding a human baseline, reviewers remained unconvinced that performance gaps on these digitized tests can be cleanly interpreted as failures of “human-like visual cognition.” The mapping from classical cognitive tests to model evaluation is interesting but indirect, and it is unclear how much the observed failures reflect limitations of language-mediated testing rather than fundamental visual reasoning deficits.

Also, AC and reviewers concern the practical impact and longevity of the benchmark. Although the authors argue that difficulty scaling and parametric generation ensure long-term value, the benchmark still depends heavily on carefully curated tasks and evaluation protocols, and it is not clear how widely it will be adopted or how much it will influence model development beyond confirming known weaknesses.

Some citations, such as this one YI Petrov. Memory structure as a psychic function. Voprosi Psikhologii, 16:132-136, 1970. does exist but even Google Scholar do not have direct link to its text, hard to verify the referenced content.

The reviewers’ concerns about incremental contribution, questionable framing, and limited new insight led me to recommend rejection.

**Reviewer Scores:**

m6KY score should be taken less weight as the material mentioned is very general, without detailed content but a plain strong positive score.

---

### Decision · Program_Chairs · 2026-01-26

Reject